# WavePulse: Real-time Content Analytics of Radio Livestreams

## ABSTRACT

Radio remains a pervasive medium for mass information dissemination, with AM/FM stations reaching more Americans than either smartphone-based social networking or live television. Increasingly, radio broadcasts are also streamed online and accessed over the Internet. We present WavePulse, a framework that records, documents, and analyzes radio content in real-time. While our framework is generally applicable, we showcase the efficacy of WavePulse in a collaborative project with a team of political scientists focusing on the 2024 Presidential Elections. We use WavePulse to monitor livestreams of 396 news radio stations over a period of three months, processing close to 500,000 hours of audio streams. These streams were converted into time-stamped, diarized transcripts and analyzed to track answer key political science questions at both the national and state levels. Our analysis revealed how local issues interacted with national trends, providing insights into information flow. Our results demonstrate WavePulse's efficacy in capturing and analyzing content from radio livestreams sourced from the Web.

## CCS CONCEPTS

• **General and reference** → **Measurement**; • **Information systems** → **Multimedia streaming**; • **Computing methodologies** → **Information extraction**.

## KEYWORDS

Web content analytics, Radio livestreams, Large language models

**ACM Reference Format:**
Anonymous Author(s). 2024. WavePulse: Real-time Content Analytics of Radio Livestreams. In *Proceedings of The Web Conference (WWW'25)*. ACM, New York, NY, USA, 21 pages. https://doi.org/10.1145/nnnnnnn.nnnnnnn

## 1 INTRODUCTION

Despite the rise of the World Wide Web and the emergence of social media networks, radio as a cornerstone of mass media has demonstrated remarkable staying power. Since 2018, even though television viewership and print readership have plummeted by 29%, radio listenership has experienced a mere 7% decrease [32]. This resilience is further underscored by radio's dominance in terms of its reach among the public. In 2023, AM/FM radio can be freely accessed by over 84% of U.S. adults, outperforming both smartphone-based social networking (78%) and live TV (72%) [21, 32].

Radio's enduring relevance stems from its unique attributes. In contrast to global social media platforms, radio's focus is primarily *hyperlocal*, and fosters deep community connections through content tailored to specific geographical areas (such as towns, counties, and states). Radio's primary function is as a *one-way* communication channel, allowing listeners to passively engage during their everyday activities, such as during commutes and/or at work. Many radio broadcasts are *spontaneous and ephemeral*, and the irreversible nature of radio broadcasts lends authenticity and immediacy to its content, particularly crucial in political discourse. These features have positioned radio as a trusted, community-oriented medium which provides an alternative to the deluge of social media content, and gives (to some) welcome respite during digital fatigue.

While these distinctions make radio unique as a medium, they also make radio content much more challenging to monitor. In the United States, these features take on heightened significance. Radio serves as a vital link across diverse urban and rural landscapes, functioning as primary information sources in remote areas and during long drives.

Media exposure, especially through radio, plays a crucial role in shaping political attitudes by both reinforcing and challenging existing beliefs. Theories of opinion formation suggest competing media messages intensity is crucial in explaining changes in opinions over time [10, 34]. While partisan news tends to modestly reinforce existing beliefs, it is also shown to activate and convert individuals when they are continually exposed to opposing viewpoints, leading them to shift away from their original affiliations and preferences [11]. Resistance to these opposing viewpoints requires the ability and motivation to recognize discrepancies between the message and one's values and beliefs [34].

The deregulation and livestreaming of talk radio content over the Web have contributed to its rise and the corresponding increase in conservative public opinion [6]. Researchers have found that exposure to talk radio can be more powerful predictor of attitudes than political knowledge [18] with information received via news radio strongly influencing policy beliefs [31] and promoting memory-based political information processing more effectively than entertainment media [17].

This paper introduces WavePulse, an end-to-end system for real-time acquisition, transcription, speaker diarization, curation, and content analysis of up to several hundreds of radio livestreams sourced from URL's accessible via the Web. The key feature of WavePulse is that most of the system components are built using powerful AI tools such as modern, multimodal large language models (LLMs) which have witnessed significant advances in 2024. This feature enables rapid design and deployment, and showcases AI's potential as a valuable tool for worldwide broadcast media analysis.

We report our findings from a pilot deployment of WavePulse that encompassed 396 AM/FM radio streams spanning all 50 US states, where we placed a strong emphasis on political news broadcasts recorded continuously over a 100-day period from late June to late September 2024. This time period captures a significant period

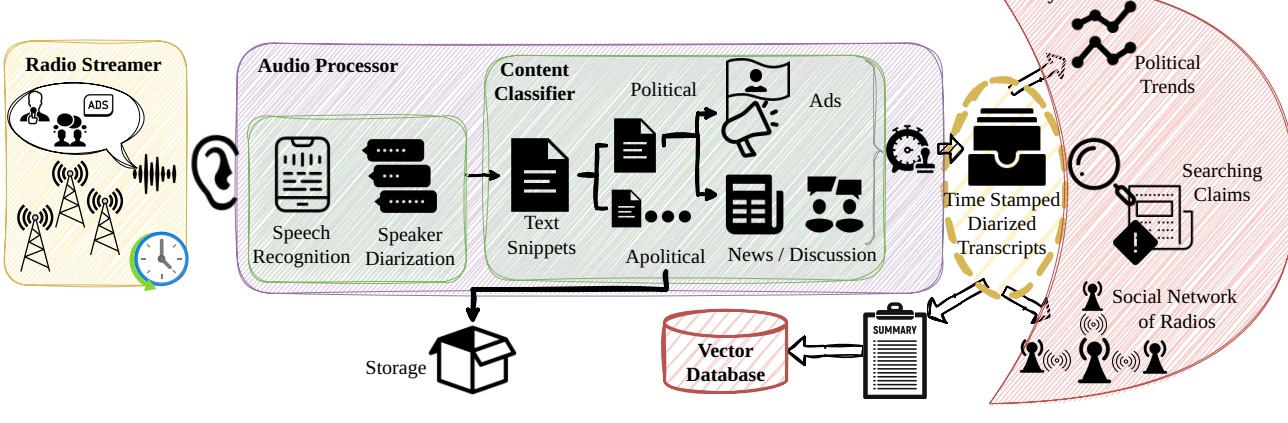

**Figure 1: Overview of WavePulse. It streams radio, transcribes, diarizes, classifies, timestamps and summarizes content on the radio, making available for analytics. We derive political trends, match claims**

of American political history, which included pivotal events such as controversial debates, two assassination attempts on a presidential candidate, an unexpected campaign withdrawal by the sitting US President, and several other key political milestones.

A key outcome of our pilot deployment of WavePulse is a large-scale, timestamped, speaker-diarized dataset of raw radio transcripts. This dataset provides a rich and unique glimpse on the pulse of the American public (as recorded on the airwaves) during this period. The corpus of transcripts was derived from over 485,090 hours of speech content,[1] comprising of 329 million text segments (1-3 sentences each) or approximately 4.5 billion words (in comparison, it matches size of the English Wikipedia as of Oct 13th, 2024 [33])[2]. We intend to publicly release this dataset for review and follow-up work by the research community. We envision that our system (and the associated dataset) can be leveraged by diverse researchers who are interested in analyzing patterns in national public discourse, media narratives, formation of public opinions, and the science of misinformation.

To showcase WavePulse, we present three case studies:

(1) **Monitoring narratives and rumors**: In our first case study, we collaborated with a team of political scientists studying election integrity and trying to identify the provenance of a very specific rumor concerning the legitimacy of the 2020 Presidential Election (which continues to echo through the political discourse 4 years later). Our system enabled us to identify positive matches for this rumor and track it across the US over a period of several months.

(2) **Understanding content syndication patterns**: In our second case study, we study content syndication across geographically dispersed radio stations. Using techniques from transcript deduplication and hashing-based matching, we construct a virtual "radio syndication graph" and find communities/clusters in this graph that frequently mirror each others' (sometimes even

niche) content. Such tools can potentially be used to study nationwide media diversity and analyze longitudinal information spread.

(3) **Measuring political trends**: In our third case study, we perform NLP-based sentiment analysis of chunks of transcripts related to specific candidates in the US Presidential Election, curate them into scalar time series, and visualize national and state-wise trends over given time periods. Remarkably, we find that our sentiment scores (gathered in a purely passive manner) mirror national polling trends, showing that WavePulse can be used as a supplementary tool for tracking public opinion over the web.

The above showcase applications illustrate the utility of WavePulse as a system for curating and comprehending content broadcast over radio. It also illuminates a specific corner of the Web (livestreamed audio) which has been remained somewhat hard to access, until the development of modern multimodal LLMs.

To summarize, our contributions include the following:

(1) An end-to-end framework for recording, transcribing and performing analytics on the radio.

(2) A data pipeline to convert raw transcripts into their rich counterparts, by time-stamping, diarizing, summarizing and classifying into ads, news and discussion, and ancilliary content.

(3) Rich analysis including topic modeling to distill top emerging narratives, sentiment analysis to gauge political temperament across the United states stratified by state and time interval.

(4) Three case studies, stemming from a collaboration with a non-profit center for monitoring election integrity.

(5) A self-updating interactive website for our analytics.[3]

The rest of this paper is organized as follows. We first describe the WavePulse framework, followed by details about the data acquisition process. Next, we provide our qualitative and quantitative findings from the three case studies described above. We provide a discussion of related work, and conclude with potential directions of future research.

---

[1]This is equivalent to 55 years of continuous speech.
[2]We acknowledge that this comparison requires careful interpretation due to the nature of spoken language and content repetition due to syndicated broadcasts.

[3]Our anonymized website can be accessed at https://wave-pulse.io.

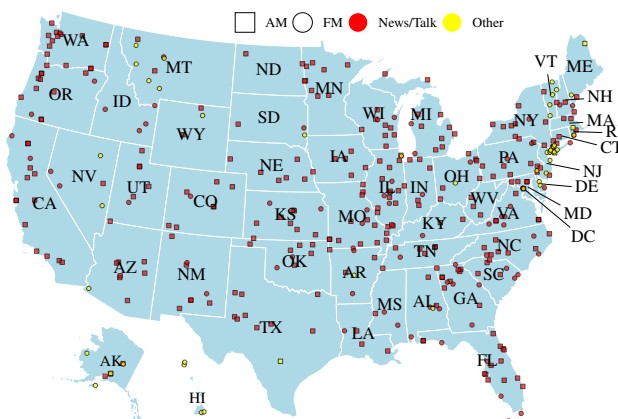

**Figure 2: Coverage of Radio Stations. Each marker is an AM / FM station. We clubbed `News/Talk/Business-News` into "News/Talk", and `Public-Radio/College/Religious/Others` into "Other". Counts: "News / Talk" : 347, "Other": 49. For rest of the US plots, we will use above state labels as reference.**

## 2 FRAMEWORK AND DATA COLLECTION

### 2.1 Design of WavePulse

The proposed framework comprises three primary components, as illustrated in Fig. 1. Each component serves a distinct function in the process of capturing, processing, and analyzing radio content:

**Radio Streamer** is responsible for acquiring audio feeds from web-based radio broadcasts [1]. It operates on a configurable schedule, enabling parallel recording of multiple audio streams at predetermined intervals throughout the day. The streamer segments incoming audio into manageable chunks to facilitate batch processing. Upon completion of each chunk, the component transfers the file to the audio buffer of the subsequent component.

The Radio Streamer continuously records all configured radio streams in parallel and segments them into 30-minute MP3 files. These files are then forwarded to the system's audio buffers for further processing. To optimize capture of relevant content while allowing time for system maintenance and backup, the streamer automatically initiates operations at 05:00 and concludes at 03:00 the following day (UTC-4).

**Audio Processor** transforms the recorded audio chunks into time-stamped, diarized transcripts through a multi-stage process:

*Diarization and Transcription:* We first utilize WhisperX [4], which integrates OpenAI's Whisper-large-v3 [24] model with PyAnnotate [22] for speaker diarization; this converts each audio files into structured JSON format. The resulting output contains spoken text segments, speaker indices, and precise start and end times for each segment (typically a sentence long, see Fig. 3 for examples).

*Content Classification:* Radio broadcasts intermix political news and discussion with ads and apolitical content. We process the radio broadcast in the JSON output using Google's Gemini-1.5-Flash model [29], which categorizes each segment as either political

or apolitical. In alignment with the project's focus on political discourse, apolitical segments are archived in cold storage.

*Advertisement Identification:* Political segments undergo a second round of classification using Gemini to distinguish advertisements from substantive content. The remaining material consists of news reports and political discussions.

Having labeled each segment as apolitical, political ad, or political content (implicitly news and discussions), we split each JSON transcript into three mutually-exclusive parts. The filtered political content is then sent for final processing.

While we started with classifying audio to segment out music and delete segments that were devoid of speech, we ended up removing this step because music was rare in news-oriented stations and radio stations tend to keep any gaps to a minimum in order to not waste air-time.

*Final Transcript Generation:* The system generates timestamped transcripts that include speaker indices using the start time of each transcript, offset with the segment-specific stamp (as illustrated in Fig. 3). Additionally, we split the transcript into three mutually-exclusive parts – news/discussion, ads, apolitical – and append continuation markers in these transcripts to prevent temporal discontinuities. For example, we insert "political ad..." between two segments of a political discussion. For more details, see Sec. A.1.

### 2.2 A Dataset of Nationwide Radio Transcripts

WAVEPULSE produces a comprehensive, segmented record of radio content, categorized into mutually exclusive, chronologically ordered, speaker-tagged, chat-like transcripts. This approach preserves the temporal integrity of the original broadcast, clearly delineating transitions between political discourse, advertisements, and apolitical content. Consequently, users can navigate the transcribed content with a clear understanding of its structure and context, even when encountering interruptions such as advertisements within political discussions. Please refer to Sec. A for details.

We collected the dataset for a period of 100 days starting June 26th, 2024 with a cutoff on Oct 3rd, 2024. In this period, we started with 158 `News/Talk` stations and scaled up to 396 stations to get wide coverage, over a course of four weeks to include stations with `News`, `Religious`, `Public-Radio`, `Business-News`, and `College` formats. Fig. 2 illustrates the coverage of 396 radio stations.

The dataset comprises of 485,090.5 hours of speech recordings, which resulted in 970,181 raw JSON transcripts. This data had approximately 4.5 billion words and 329 million text segments, each are 1-3 sentences long.

**Quality of Transcripts.** Each recording goes through several steps before being converted into the final transcripts. Out of these steps, speech recognition is the most important as we discard audio recordings after successful transcription, making this step irreversible.

In order to make evaluation of downstream baselines feasible, we filtered a representative dataset comprising two weeks worth of recordings, i.e., totaling 672. We varied time-of-day, U.S. state of station, format, wave modulation (AM/FM). We also ran checks to ensure that no filtered recording was cutoff and held a full 30 minute worth of speech. We evaluated Nvidia's RNN-T-Parakeet-1.1B [20], MMS-1B [23], WhisperX [4], while considering Microsoft

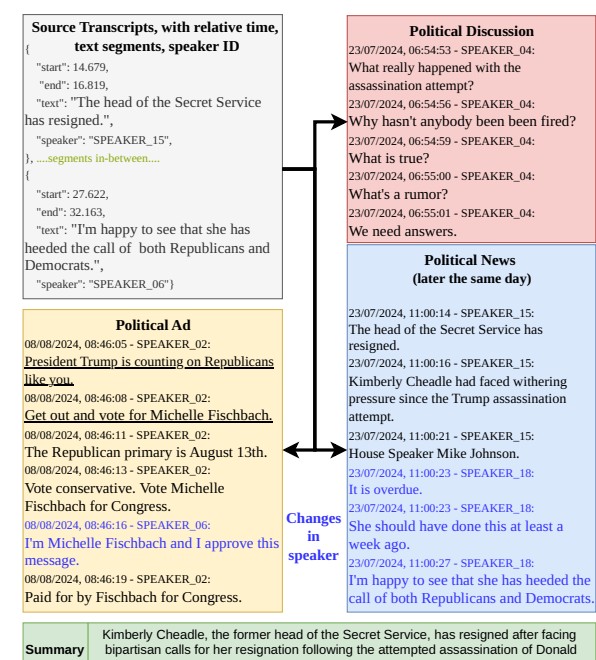

**Figure 3: Samples of** (Top left) **JSON segments** (Bottom Right) **Corresponding Diarized Time-stamped Political News** (Top right) **Discussion**, (Bottom Left) **Advert.**, and (Bottom) **Summary**.

Azure as the ground truth on this dataset. WhisperX performed most accurately and fastest (see Tab. 1).

**Condensing the dataset.** Radio discussions and news reporting are essentially conversations. Each radio station presents the news selectively, brings guests to discuss them, and broadcasts their opinions to listeners. To create concise summaries of each conversation, preserving relevant discussion and news threads while reducing erroneous text predictions, we summarize each 30 minute transcript using Gemini [29]. This step also enhances suitability of our dataset for open research by filtering out personally identifiable information. Fig 3 illustrates a sample summary.

**Embedding the dataset.** As the dataset has nearly a million data points, it warranted an efficient search mechanism. Therefore, we converted the summarized version of our dataset into a 1024-dimensional vector database using dense embeddings from BGE-M3 [9]. Each embedding vector contains metadata like state, call sign, date, and time. We used LLaMa-3.1-8B-Instruct [14] to query the database, which in turn used FAISS [12] to search through the vector space and retrieve top matches. This resulted in a Question-Answering Retrieval Augmented Generation (QA-RAG) pipeline.

## 3 ANALYSIS AND CASE STUDIES

Having a time-stamped radio transcript database, that captures a unique snapshot in American political broadcasting, we analyze them in the following three case studies. The studies demonstrate

**Table 1: Word-error-rate and Avg. Inference Speed for 30-min audio clips of ASR models, from our representative dataset.**

| Model | RNN-T | MMS-1B | WhisperX |
|---|---|---|---|
| WER (%) ↓ | $14.5^{\pm 8.2}$ | $35.1^{\pm 13.2}$ | $\mathbf{8.4^{\pm 4.6}}$ |
| Speed (s)↓ | 15.0 | 17.8 | **9.5** |

how WAVEPULSE can be used to explore the spread of a specific misinformation claim, the amplification of information by a network of radio stations, and assessment of the overall sentiment of major party candidates.

### 3.1 Case Study: Spread of a Political Narrative

**Overview.** We collaborated with a democracy group at a non-profit center which champions social causes including election integrity. Our goal was to understand how a system like WAVEPULSE could be useful to gain insights into the political/election discourse.

The center aimed to track a narrative that revolved around the integrity of the 2020 US Presidential election in Fulton County, Georgia (US), that stemmed from a report analyzing the election in Georgia, published by a campaign spokesperson, claiming that the election was stolen from Trump. Taking this narrative as an example, we searched through our corpus for matching pieces of the narrative. Our dataset came out positive with at least 50 positive samples, including a majority amplifying the claim in this narrative, a handful reporting and a few debunking it.

**Context and Background.** A major focus area of the democracy group at this non-profit is to perform election monitoring worldwide. A specific current goal of theirs is to track election related claims across the U.S. Prior to *WavePulse*, their reach was limited in scope to social media posts, blogs, news articles, podcasts and comments. Political scientists at the center provided us with a narrative that appeared on the airwaves just after the Biden-Trump's debate on June 27th, 2024; this date coincided with the start of our data collection. The main rhetoric was that there were inconsistencies in the logic and accuracy tests conducted on voting machines.

> **Summarized narrative:** "There were discrepancies in the 2020 Presidential election vote count in Fulton County, Georgia. Specifically, 17,000 votes needed to be reconciled before certification. A group called *The Elections Group* was involved in various aspects of the 2020 election across multiple states. The Georgia State Election Board investigated the recount process and confirmed some rule violations by Fulton County. Over 20,000 ballots were added to both the original results and the machine recount without proper justification. There were missing ballot images and duplicate ballots, which makes the election suspicious. The Secretary of State's office and the State Election Board investigated, and there was a lack of transparency and accountability. Since the 2020 election in Georgia was inaccurate, therefore the Presidential election was stolen."

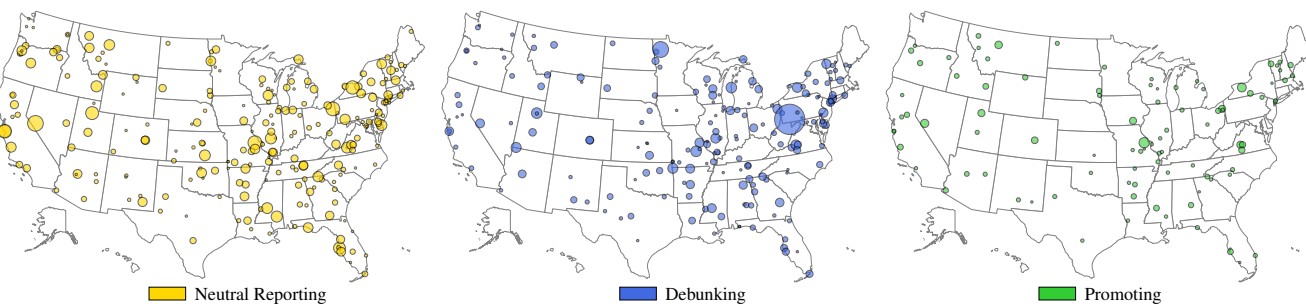

**Figure 4: Occurrence of Neutral Reporting (51.0%), Debunking (36.3%) and Promoting broadcasts (10.4%) related to the 2020 Election narrative (2.3% were unknowns). We encoded number mentions in the size of bubbles and use identical scale throughout.**

They also provided us with a list of keywords, which we converted to rules for filtering transcripts:

```
 logic AND accuracy AND test       OR Logic and accuracy
OR logic test* AND election        OR L and A
OR logic testing AND vot*          OR logic/accuracy
OR accuracy testing AND vot*       OR logic / accuracy
OR accuracy test* AND election     OR LA testing
OR voting machine test*            OR Elections group
OR machine test AND election       OR <Claimant's name>
OR machine test AND vot*           OR <Website's name>
```

**Manual Methodology:** Using the above keywords, we first filtered transcripts of stations in Georgia. E.g., a filtered transcript should have "logic" AND "accuracy" AND "test" keywords. However, this resulted in only a few matches. We expanded our search nation-wide, which revealed 120 samples. As keywords acted as a broad net and we needed exact citations, we did an internal human study amongst four individuals. We split the 120 samples into four overlapping sets so that each transcript gets seen atleast twice. Then, we aggregated the responses and received a ground truth with 53 transcripts – 34 supporting the claim, 16 neutrally reporting and 3 actively debunking this claim. An excerpt from a June 2024 transcript that matched:

> **A matching transcript from Ohio (US) reads:**
> "You know, here, the must-read of the day, [Claimant's name] on how Georgia was stolen. A longtime spokesperson for President Trump, published a must-read analysis of the 2020 stolen Georgia election on [website], Wednesday. This is the most comprehensive investigative piece of the fraudulent election in Georgia to date.
> Biden allegedly received the most votes of a president in U.S. history, yet simultaneously could only sneak up in Georgia by some 13,558 votes. It didn't add up. It just didn't add up, and neither did the numbers in Fulton County."

**Automated Methodology:** As the manual method depends on curated filtration rules to reduce search space, which is non-trivial for arbitrary narratives, we employ QA-RAG. We started by querying it, which gave us a positive answer:

> **Question to LLM**: Were there discrepancies in the 2020 Presidential election vote count in Fulton County, Georgia?
> **Answer**: Yes, there were claims of discrepancies which reported that Fulton County had double-counted thousands of ballots during a recount, resulting in discrepancies in the vote count. This issue was brought up by a whistleblower and led to an investigation. he Georgia Election Board voted 3-2 to request an investigation by Attorney General Chris Carr into the matter. Though, the outcome of the election was not altered by these discrepancies, but the incident raised concerns about the accuracy and reliability of the Georgia election process in 2020.

While the retrieval mechanism yielded solid leads, we did not have a ground truth to compare it against. Therefore, to get a holistic understanding of the scope of this narrative, we searched through the whole dataset. For each summarized transcript, we prompted Llama to count the number of mentions related to "2020 election being stolen, rigged, or false." If a transcript came out positive, we subsequently obtained a label for its stance, i.e., *Promoting*, *Neutral Reporting*, or *Debunking*. We obtained 3,497 mentions across the corpus. Fig. 4 illustrates the occurrence of such mentions and provides breakdown across the stances. For prompts see A.2.

**Findings and Discussion:** While the manual method yielded 53 matches, the automated method extracted 3,497 matches, providing a superior estimate of the media landscape covering a contentious narrative. Majority of mentions were examples of neutral reporting, followed by 36.3% of debunking the narrative, with only a 10.4% minority promoting it. This case study serves as an example of how one can gauge the level of traction a narrative gets on the radio.

## 3.2 Case Study: Content Syndication Across Radio Stations

Our analysis of radio station transcripts revealed extensive verbatim duplications across geographically dispersed stations, suggesting the existence of a complex social network among broadcasters. This phenomenon, observed across state boundaries and varying time frames, indicates structured information sharing among media outlets. For instance, a specific claim regarding a presidential candidate's alleged substance use before a debate was simultaneously

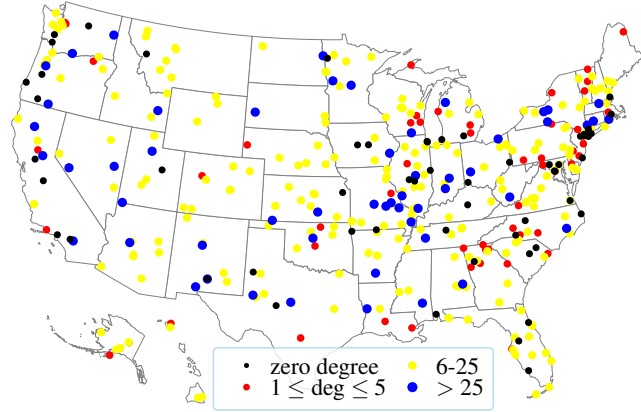

**Figure 5: The "Syndication" Social Network among Radio. We do not show edges for clarity. Here, each marker is a station and color encodes its degree category.**

broadcast by 31 distinct stations. While this synchronicity in content dissemination does not establish causality between broadcasts, it strongly suggests coordinated information sharing.

**Methodology: Connecting and Categorizing Stations.** To investigate information sharing patterns, we developed an algorithm to identify unique broadcasts and their repeats, comprising the following steps:

*Hashing and Similarity Computation:* We computed locality-sensitive hashes of text-only portions of all transcripts using MinHash. We considered transcript pairs with Jaccard similarity exceeding a predefined threshold ($\theta = 0.8$) related and thus added to each other's adjacency list. As causality is hard to predict, we consider such a content-based match to only suggest a symmetric connection.

*Subgroup Identification:* Utilizing these adjacency lists, we expanded our search to identify distinct subgroups through a Breadth-First Search (BFS) approach. We started with the initial list of unvisited transcripts, and BFS all connected transcripts, forming exhaustive lists of resonating broadcasts which matched thematically.

*Network Refinement:* To identify long-term collaborations and information propagation hubs, we implemented the following steps:

(1) Merged broadcasters in the same subgroup on consecutive dates (e.g., We would consolidate KM_WXYZ_2024_07_15_13_30 and LM_WABC_2024_07_16_02_00 broadcasted identical content).

(2) Discarded single-broadcaster subgroups, eliminating instances of content repetition on two-consecutive days.

(3) In the remaining subgroups, extracted only the station names, such as KLMN and KOPQ, for each unique station.

(4) Removed single-station lists, further refining the network by eliminating stations who broadcast their own content several days apart.

(5) Created bidirectional edges between stations in each subgroup (e.g., for stations KLMN, KOPQ, and KRST, edges were created between all pairs).

(6) Ensured uniqueness across rows and order invariance, standardizing edge representation.

(7) Generated pair-wise connections, excluding self-connections.

**Results and Discussion.** Our analysis initially identified 22,149 unique subgroups broadcasting similar content. Post-refinement, this reduced to 1,776 subgroups with 2,684 unique edges. This content mirroring pattern suggests coordinated messaging strategies transcending geographical and temporal boundaries. Figure 5 illustrates this broadcasting station network. Notable findings include:

- Fifteen stations exhibiting over 40 connections, suggesting key information exchanges.
- A ten-station network spanning 10 mid-western and southern states shared content several times, indicating a regional syndication network.
- Cross-country information flow, exemplified by KYZZ (California) and WABC (New York) broadcasting identical content with a consistent two-day delay.
- 50 stations remained disconnected in our final network, potentially indicating non-participation in syndicates, self-broadcasters or representing false negatives in our analysis. For instance, a station in New Jersey, despite being a major broadcaster, showed no connections in our network, suggesting it might prioritize original content or use syndication methods our analysis could not capture.
- Content propagation chains, such as a station in Iowa broadcasted a story, another one in Tennessee echoed it ten days later, and followed by Illinois after another eight days.

This study enhances our understanding of information propagation in legacy media networks. The observed patterns raise important questions about media diversity, centralization of narrative control by major syndicates, and the potential for rapid, wide-scale dissemination of specific viewpoints across seemingly independent broadcast radio channels. This methodology also provides a simple approach to map information flow and identifying potential echo chambers in the broadcasting landscape.

### 3.3 Case Study: Presidential Candidates' Favorability Trends

The summer of 2024 marked a pivotal period in American politics, with public perception of presidential candidates fluctuating in response to unfolding events. This study delves into these dynamics through a sentiment analysis of the dataset, focusing on the three most prominent figures: Harris, Biden, and Trump, with Biden dropping out in mid-July.

We isolated relevant text segments by keyword matching, carefully excluding instances of multiple candidate mentions to ensure sentiment clarity[4]. The Twitter-roBERTa-base model [8], denoted as $\mathcal{S}$, served as the foundation for sentiment analysis, generating positive ($S_\oplus$), neutral ($S_\odot$), and negative ($S_\ominus$) sentiment counts. To distill these multifaceted sentiment counts into a single, comprehensible metric, we developed a normalized sentiment score $\bar{S} \in [0, 1]$:

$$\bar{S} = (2 \cdot S_\oplus + 1 \cdot S_\odot + 0 \cdot S_\ominus)/(2 \cdot S_T)$$

$$\text{where } S_\oplus, S_\odot, S_\ominus \in \mathcal{Z}^+ \text{ and } S_\oplus + S_\odot + S_\ominus = S_T$$

This formulation captures the nuances of all three sentiment categories, while providing a holistic view of content sentiment.

---

[4]Name variations (e.g., "Kamala" for Harris) were aggregated under primary identifiers for consistency.

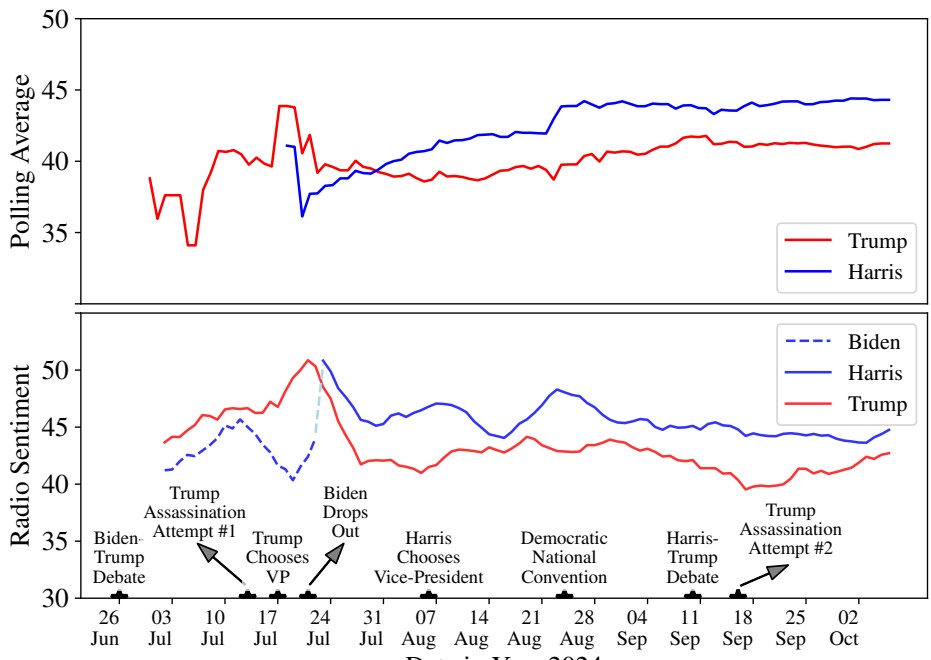

Figure 6: (Top) National Polling Averages from the Silver Bulletin blog [28] for presidential candidates Trump and Harris, starting on Jul 19, 2024 just before Biden's withdrawal. (Bottom) Normalized sentiment for political candidates obtained via the radio. The plot was averaged using a seven-day sliding window, with annotations indicating key political events. There is an anomalous peak for Harris, when Biden dropped out, because before that date there was lesser mentions about Harris, which peaked when she was brought forward as the Democratic Presidential Candidate.

The national polls are derived by soliciting the polls from the public and weighting them according to their influence, while the sentiment plot reflects the temperament of the radio content. We observe similar trends in both, suggesting that the radio content can be used as a proxy of its listeners' sentiment.

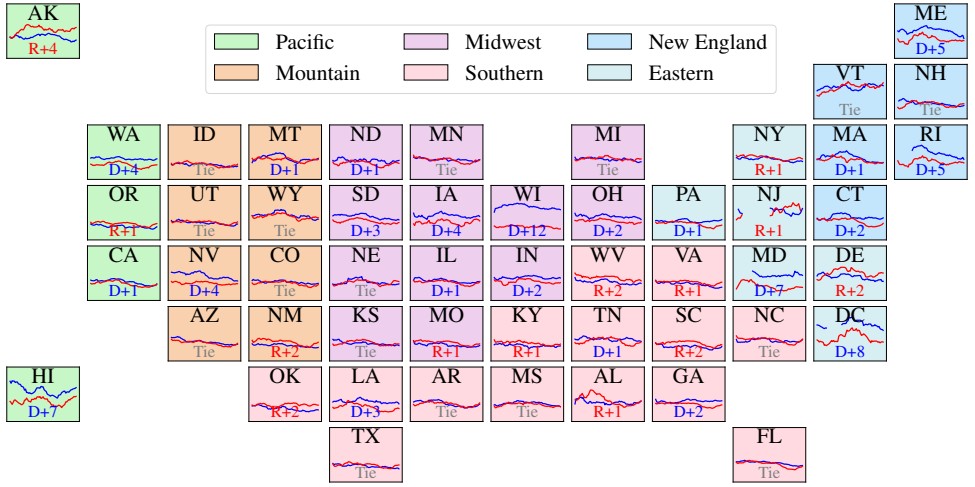

Figure 7: State-wise normalized sentiment towards Trump (red) and Harris (blue) in the period of Jul 21st - Oct 3rd, 2024 on a 14-day rolling average. Each state is annotated with either Republicans (R) or Democrats (D), along with a number which represents mean percentage gain of one party candidate over another throughout the considered period. If gain was < 1, we label it as a Tie. New Jersey (NJ) had a data anomaly because there were only a few stations which streamed online did not have restrictive terms-of-service.

The lower part of Fig. 6 illustrates the ebb and flow of nationwide sentiment as computed from the radio content, smoothed with a 7-day moving average to reveal underlying trends, as radio shows had less programming during weekends which caused weekly dips. Annotated political events offer context for significant shifts, painting a picture of how key moments shaped public perception. We derive the upper part of the figure from raw data of Nate Silver's model [28].[5] Diving deeper, Fig. 7 breaks the trends down according to state-specific sentiments, reflecting local political climates or the impact of targeted campaign strategies.

**Findings and Discussion:** Our sentiment predictions demonstrate similarity with the 2024 Presidential polling averages, which in turn is based on reputable national polls and summarized by a competitive model from a prominent pollster. This alignment suggests that radio content analysis can serve as a valuable proxy for public sentiment, offering real-time insights into political trends.

The state-wise sentiment analysis reveals a granular view of political leanings across the country. However, some anomalies emerged, such as the surprisingly strong Democratic lead in Wisconsin (D+12). This discrepancy between state-level and nationwide trends warrants further investigation.

## 4 RELATED WORK

**Radio Content Analytics.** While radio has a century-long history as a broadcasting medium for entertainment and information dissemination, modern radio in the U.S. has its roots in the deregulation adopted in the Telecommunications Act of 1996 that fundamentally reshaped the U.S. radio industry. The deregulation altered the industry's economics, with large conglomerates implementing cost-cutting measures such as staff reductions and automated programming, while also changing advertising dynamics by offering multi-station, multi-market packages to advertisers [13]. Also, due to the rise in online music streaming and piracy making music expensive to broadcast, talk shows gained popularity. We did not include iHeartMedia stations in this study as they have restrictive terms of service, but still found several other syndicates [6].

Hofstetter [15, 16] studied how radio shows shape public opinion and found that they play several roles for their listeners, including seeking information, contextualizing, interpreting the information, and serving as a proxy for interaction with the hosts and guests.

From 2006-2011, DARPA undertook efforts to collect and transcribe cross-lingual broadcast news and talk shows under its GALE project [30]. In 2019, RadioTalk [5] was the first work that created a large corpus of talk radio transcripts comprising 284,000 hours of radio and 2.8 billion words. The authors conducted transcription using a TDNN model which produced noisy samples with a WER of 13.1%. Using this dataset, Brannon and Roy [7] compared the speed of news on Twitter versus radio during 2019-2021 and found that Twitter news circulates and evaporates faster and is more negative than radio. A follow-up work assembled the Interview media dialog dataset [19] comprising of collection of 20 years of NPR radio transcripts that enables discourse pattern analysis. Our work simultaneously provides an end-to-end pipeline, based on modern

LLMs with 8.3% WER, which continuously produces a rich dataset while being able to run real-time analytics to poll it.

**Social Media Analytics Frameworks.** Aggarwal et al. [2] developed a multimodal framework to track bias and incivility on Indian TV news. Saez-Trumper et al. [27] used unsupervised methods on a geographically diverse set of news sources by examining 'gate-keeping,' coverage, and statement bias to find bias in online news. Ribeiro et al. [26] employed scalable methodologies that leverage social media's advertiser interfaces to infer the ideological slant of thousands of news outlets. Allen et al. [3] analyzed Facebook posts during the COVID pandemic for content in the grey area and found that this unflagged content cast doubts on vaccine safety or efficacy and was 46-fold more consequential for driving vaccine hesitancy than flagged misinformation. Contrary to them, WAVEPULSE provides a tool to measure radio content in real-time, with the case studies performed primarily to showcase its capabilities, and it does not provide any subjective labeling.

## 5 DISCUSSION AND CONCLUSION

**Ethics statement.** We have maintained strict ethical principles during the data collection, usage, and analysis conducted in this work. Our research utilizes data broadcast to the web on public radio streams, which falls under fair use unless explicitly restricted under terms of service. We meticulously reviewed broadcasters' license agreements where applicable and excluded stations with such restrictions. The dataset does not contain personally identifying information (PII) about listeners. The dataset may include PII about advertisers (e.g., names and contact information of organizational representatives) and show hosts. Given our focus on political topics, we have strived to avoid political bias. We acknowledge that the usage of LLMs in our analysis may inherently exhibit some biases due to their respective training data.

**Limitations and Future Directions.** Our analysis does not incorporate population data along with reach of each stations waves' to calculate exposure to each station. Nielsen Audio sells exposure ratings and FCC hosts ground conductivity data. We also included only stations which are livestreamed over the Internet. Terrestrial-only radio broadcasts would require dedicated hardware (an antenna, transceiver, and recording equipment). Finally, while WAVEPULSE is widely applicable, our analysis derives results and conclusions from only US radios. An important direction of future work is to broaden the scope to worldwide radio livestreams; due to the multilingual nature of LLMs we anticipate our system to scale up with no significant design changes.

**Conclusions.** We introduce WAVEPULSE, an end-to-end pipeline for gathering and analyzing live-stream radio broadcasts which can increasingly be accessed via the Web. Using this system, we collected nearly half a million hours of news/talk radio content over a 100-day period of significant political activity in the United States. We conducted three case studies: tracking political narratives with political scientists, building a social network of radio stations, and predicting political trends in real-time. Our findings highlight the depth of insights derivable from *WavePulse*'s comprehensive dataset.

---

[5]Nate Silver is a renowned American statistician and data journalist, famous for accurate election predictions and founding the influential website FiveThirtyEight. This website focuses on opinion poll analysis, politics, and economics. Their data (accessed Oct 13) is displayed in the upper part of Fig 6.

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

# A SUPPLEMENTARY MATERIAL FOR WAVEPULSE

Table of Contents:

## A.1 Data Collection Pipeline

*A.1.1 Radio Streamer.* This component is used to stream and record audio streams from multiple radio broadcasts based in parallel. It takes as input a configurable schedule in the form of a json file, for when to stream and from which radio. There will be one entry per radio stream, its live URL, name, record times, and the state in which it is locate along with a list start and end times for streaming for that particular radio stream as shown in Listing 1. The component processes the schedule to get unique start times across all the stations and durations specific for each station and then create cron jobs appropriately to achieve it's objective.

Conventionally, radio stations are referred by their call signs (3-4 lettered string) with the starting letter being either **W** (Stations east of the Mississippi River), **K** (Stations west of the Mississipi River), **N** (military stations), **A** (Army or Air Force stations). See Table 2, Table 3, and Table 4 for the full list of successfully streamed stations.

Audio files are recorded and saved in chunks of 30 minutes to facilitate batch transcription and analysis. The recorded audio files are distributed into buffer folders for running transcription in parallel. Buffer folders are created based on the number of available GPUs in the system to run transcription. The files are named in format *SS_RRRR_yyyy_mm_dd_HH_MM.mp*3 where SS stands for State Abbreviation, RRRR stands for radio call-sign which is unique for each radio station , followed by year, month, day, hour and minute, such as, *CA_KAHI_*2024_07_16_13_30.*mp*3. This naming format allows us to be able to easily filter files based on state, radio station or dates.

```
{
  "url": "https://stream.revma.ihrhls.com/zc3014",
  "radio_name": "KENI",
  "time": ["08:00", "14:00", "17:00", "21:30"],
  "state": "AK"
}
```

**Listing 1: Example for schedule of one radio stream. As per schedule radio streamer will record the audio from 8:00 AM to 2:00 PM then from 5:00 PM to 9:30 PM**

*A.1.2 Audio Processor.* This component is responsible for transcription, adding punctuation and capitalization to the text, providing time stamps, and speaker diarization, of the audio files recorded. We tried multiple ASR models for transcription like facebook's

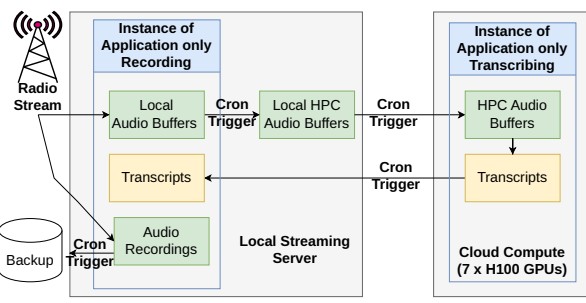

**Figure 8: Crontab Schedule in the data collection pipeline to transcribe on the cloud and retrieve back.**

MMS-1B, Nvidia's parakeet-rnnt-1.1b and OpenAI's whisper-large-v3. Fig. 2 has sample outputs of these models for same input audio. We found word error rate of mms-1b to be significantly higher as compared to parakeet-rnnt and whisper-large-v3, for the later two it was comparable. Ultimately we decided to go ahead with WhisperX implementation of Whisper as it provides a built-in pipeline for transcription using Whisper, accurate timestamps using wav2vec2 and Speaker Diarization using Pyannote at a reasonable inference speed. Listing 1. shows a snippet of output using WhisperX pipeline.

```
{
    "start": 946.93,
    "end": 948.391,
    "text": "Here's Anna with headlines.",
    "speaker": "SPEAKER_04"
},
{
    "start": 948.532,
    "end": 959.54,
    "text": "Pope Francis today has accepted the
        retirement of the longtime Archbishop of
        Boston, Cardinal Sean O'Malley and chosen
        Providence Bishop Richard Henning to be his
        successor.",
    "speaker": "SPEAKER_19"
},
{
    "start": 960.601,
    "end": 964.004,
    "text": "Tropical Storm Debbie is now a hurricane
        .",
    "speaker": "SPEAKER_19"
},
{
    "start": 964.585,
    "end": 965.045,
    "text": "What, Matt?",
    "speaker": "SPEAKER_19"
},
{
    "start": 966.766,
    "end": 968.087,
    "text": "Oh, no, my mic is still alive.",
    "speaker": "SPEAKER_04"
}
```

**Listing 2: Sample Transcript Segments**

One H100 GPU takes around 30 seconds to process one audio file of 30 minutes. So one GPU can process 60 audio files of 30 minute length, in other words one GPU can process 60 radio streams without resulting in any backlog. Since we were trying to process 400+ streams so we had to use 7 H100 GPUs. Due to resource

constraints in our local server we used cloud services to get access to GPUs to be able to process all the radio streams. One instance of the application runs on local server and it's job is to only record the streams and store audio files in a recordings folder and their copies to audio buffer folder. Normally, files in recordings folder are sent to backup regularly and the ones in audio buffer are used for transcription and then deleted. In this case we setup a corn trigger to periodically transfer files from audio buffer folders to hpc audio buffer folders with load distribution. Then next cron job periodically transfers files from hpc audio buffer folders to the audio buffer folders of cloud compute service. We have another instance of application running on cloud server with 7 H100 GPUs that does only transcription part and saves the result in transcripts folder. Then another cron trigger transfers files from the transcripts folder on cloud server to local server. If the audio streams are less and can be processed by local server than one can enable both recording and transcription on local server without needing to use cloud resources and creating data pipelines.

Publish it on Amazon. Put yourself out there. If there's something that you, if there's advice you want to give, put it out there.

**(a) WhisperX [4]**

publish it on amazon put yourself out there if there's something that you if there are if there's advice you want to give put it out there

**(b) RNN-T [25]**

publis it on amazo put ourself out there if ther is something a tere is adie ou want to giv out

**(c) MMS-1B [23]**

**Figure 11: Transcribed text for same audio clip using different speech recognition models.**

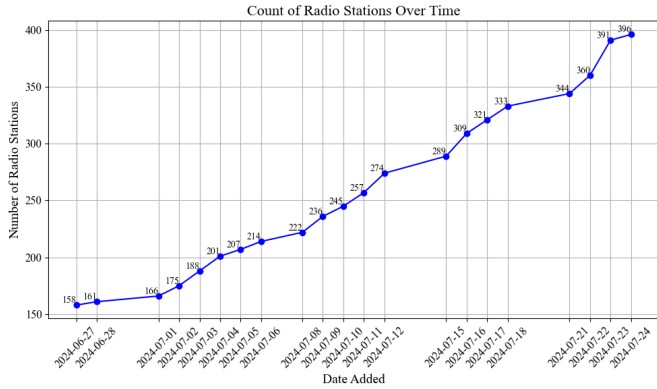

**Figure 9: Line chart showing the addition of radio stations over time.**

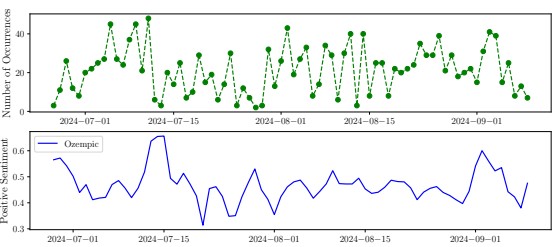

**Figure 12: Sentiment for Ozempic, a pharmaceutical product which has been gaining traction due to success in fat loss. This plot suggests that talk shows discuss a variety of things and not just politics.**

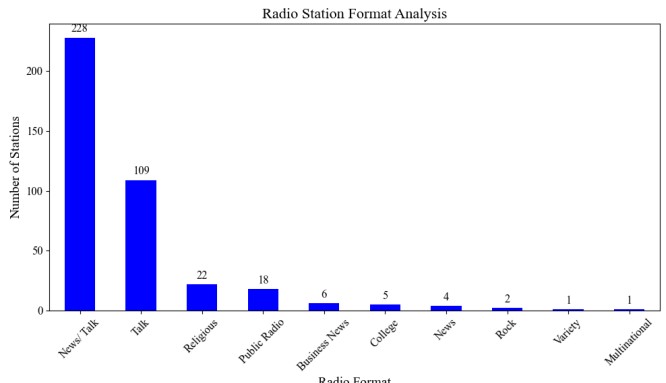

**Figure 10: Bar chart showing the number of radio stations by station format.**

## A.2 Summarization

The summarization process is a critical step in condensing lengthy conversation transcripts into concise, meaningful summaries. This section outlines the prompts and techniques employed to achieve efficient and accurate summarization using the Google Gemini API, combined with dynamic conversation segmentation and embedding generation. Our unit of summarization is a 30-min transcript.

**Summarization Prompt:** You are a concise and direct news summarizer. Given below is a JSON with spoken text and its speaker ID recorded from a radio livestream. Create a summary that:

- Presents information directly, without phrases like "I heard" or "The news reported."
- Uses a factual, journalistic tone as if directly reporting the news.
- Retains key facts and information while making the content specific and granular.
- Removes personal identifiable information (PII), such as phone numbers and sensitive personal data, but keeps public figures' names (e.g., politicians, celebrities) and other key proper nouns relevant to the context.
- Is clear and avoids vague language.
- Clarifies ambiguous words or phrases.
- Utilizes changes in speaker ID to understand the flow of conversation or different segments of news.
- Corresponds strictly to information derived from the provided text.
- Organizes information into coherent paragraphs, each focusing on a distinct topic or news item.
- Maintains a neutral, objective tone throughout the summary.

Do not include any meta-commentary about the summarization process or the source of the information.
Spoken Text Transcription: {conversation block}

---

**Prompt used for finding matching mentions/claims of the 2020 election narrative:** Analyze the following document summary regarding mentions of the 2020 election being stolen, rigged, or false.
Document summary: {content}
Answer the following questions:
- How many times was the 2020 election being stolen, rigged, or false mentioned?
- Did the document promoting, neutral report, or debunk these claims?
Provide your answer in the following format:
"mention_count": <number of mentions>,
"stance": "<promote/neutral/debunk>"

## A.3 Dense Embedding for Summaries

*A.3.1 Overview.* Once the conversation segments are generated and summarized, the next step is to create embeddings for each summary. These embeddings serve as high-dimensional vector representations that capture the semantic meaning of the text. The embeddings are generated using BGE-M3 [9]. By converting textual summaries into vectors, we enable efficient operations like similarity comparisons, clustering, and semantic search.

*A.3.2 Mathematical Representation.* The embedding function, denoted by $f$, transforms a given summary $S$ into a high-dimensional

vector $\mathbf{v} \in \mathbb{R}^d$, where $d$ represents the number of dimensions in the vector space. The process can be mathematically described as:

$$\mathbf{v}_S = f(S), \quad f : \mathbb{R}^n \to \mathbb{R}^d \quad (1)$$

Here, $S$ is the summary, $\mathbf{v}_S$ is its corresponding embedding, $n$ is the number of words in the summary, and $d$ is the dimensionality of the embedding space. This transformation allows each summary to be represented as a vector, which can then be compared against other vectors in terms of cosine similarity or other distance measures.

*A.3.3 Cosine Similarity of Embeddings.* To compare the semantic similarity between two summary embeddings, cosine similarity is used. The cosine similarity between two vectors $\mathbf{v}_i$ and $\mathbf{v}_j$, representing summaries $S_i$ and $S_j$, is given by:

$$\text{Cosine Similarity}(\mathbf{v}_i, \mathbf{v}_j) = \frac{\mathbf{v}_i \cdot \mathbf{v}_j}{\|\mathbf{v}_i\|\|\mathbf{v}_j\|} \quad (2)$$

Where $\mathbf{v}_i \cdot \mathbf{v}_j$ represents the dot product of the vectors, and $\|\mathbf{v}_i\|$ and $\|\mathbf{v}_j\|$ are the magnitudes (or norms) of the vectors. This similarity measure is particularly useful for clustering, retrieval, and semantic search tasks.

*A.3.4 Asynchronous Embedding Generation.* Given the large number of summaries, the embeddings are generated asynchronously to improve performance and scalability. By parallelizing the generation process, we can significantly reduce the processing time, ensuring that embeddings are computed efficiently for each summary.

The embedding function is queried asynchronously for each summary, as shown in the following pseudo-code:

By leveraging the embeddings, we ensure that each summary is represented in a vector space, allowing for more advanced semantic operations.

*A.3.5 Usage of Embeddings.* The generated embeddings are then used for multiple downstream tasks, such as:

- **Clustering**: Grouping similar conversations based on their semantic embeddings.
- **Retrieval**: Efficiently finding summaries that are similar to a given query.
- **Semantic Search**: Searching through conversation summaries based on the semantic content rather than exact matches.

## A.4 Semantic Similarity Search with FAISS

*A.4.1 Why FAISS?.* FAISS (Facebook AI Similarity Search) was chosen for its ability to perform efficient nearest-neighbor search in high-dimensional vector spaces. Given the large number of summaries and their associated embeddings, FAISS provides a highly scalable solution for searching through these embeddings.

*A.4.2 FAISS Workflow.* The FAISS workflow consists of the following steps:

(1) Initialize a FAISS index with the appropriate dimension size $d$, where $d$ is the size of the embedding vectors.
(2) Add the embeddings for each summary to the index.

(3) Perform nearest-neighbor search queries on the index to find semantically similar summaries.

The FAISS index is initialized as:

$$\text{FAISS Index} = \text{faiss.IndexFlatL2}(d) \qquad (3)$$

Where $d$ is the dimension of the embedding vectors. The similarity search is then performed by querying the index with the query embedding $q$.

$$\text{Distances, Indices} = \text{FAISS Index}.search(q, k) \qquad (4)$$

Where $k$ is the number of nearest neighbors to retrieve.

## A.5 Useful Addition: Finding Unique Narrative Network

Analysis of transcripts revealed that the same narrative, content, or show was often broadcast multiple times on a radio station and across multiple stations, even when the stations appeared independent. An algorithm was developed to identify unique narratives that were shared multiple times and form a network of radio stations for each narrative. MinHash with a similarity threshold of 0.8 was used to check if any two transcripts contained the same content. This process identified approximately 22,000 unique narratives from the dataset. *We use only a part of it in WavePulse, where we perform LSH and BFS to find a social network. The narratives from this subsection can help us answer: What is getting amplified in that network?*

> **Output Summary of Common Content:** This collection of text excerpts focuses on the upcoming US presidential election and the role of celebrities in influencing voters.
> **Election Coverage:**
> - The Democratic National Convention concluded with Vice President Kamala Harris accepting the party's nomination for president.
> - The convention featured speeches from prominent Democrats and celebrities, highlighting their support for the Harris-Biden ticket.
> - The election is just two months away, with the first presidential debate scheduled for September 10th.
> **Celebrity Endorsements:**
> - Celebrities are increasingly using their platforms to endorse political candidates.
> - While some celebrities, like Oprah Winfrey, have been shown to have a significant impact on voter turnout, others, like Taylor Swift, have not yet endorsed anyone.
> - The rise of social media and AI-generated images has created new challenges for verifying the authenticity of celebrity endorsements.
> **Other Topics:**
> - The text also includes information on the economy, specifically the role of data-driven decision-making in various industries.
> - The text also touches on the importance of free speech and the potential risks of "cancel culture."

---

**Algorithm 1** Finding unique narratives in broadcasts

---

**Require:** Set of radio broadcast transcripts $T = \{t_1, t_2, ..., t_n\}$
**Require:** Similarity threshold $\theta$
**Require:** Time threshold $\Delta t$
1: **function** OPTIMIZEDANALYZEBROADCASTS($T, \theta, \Delta t$)
2:     $H \leftarrow$ COMPUTELSH($T$)   ▷ Compute LSH for all transcripts
3:     $A \leftarrow$ CREATEADJACENCYLIST($T, H, \theta$)
4:     $N \leftarrow$ IDENTIFYNARRATIVESDISJOINTSET($A$)
5:     **return** $N$
6: **end function**
7: **function** CREATEADJACENCYLIST($T, H, \theta$)
8:     $A \leftarrow$ empty dictionary
9:     **for** each $t_i \in T$ **do**
10:         $C \leftarrow$ GETCANDIDATES($H, t_i$)   ▷ candidates from LSH
11:         **for** each $t_j \in C$ **do**
12:             **if** MINHASHSIMILARITY($t_i, t_j$) $\geq \theta$ **then**
13:                 $A[t_i] \leftarrow A[t_i] \cup \{t_j\}$   ▷ add to matching set
14:                 $A[t_j] \leftarrow A[t_j] \cup \{t_i\}$ ▷ maintain symmetric set
15:             **end if**
16:         **end for**
17:     **end for**
18:     **return** $A$
19: **end function**
20: **function** IDENTIFYNARRATIVESDISJOINTSET($A$)
21:     $N \leftarrow \emptyset$   ▷ Set of disjoint repeated narratives
22:     $V \leftarrow \emptyset$   ▷ Set of visited nodes
23:     **for** each $t \in A$.KEYS not in $V$ **do**
24:         $n \leftarrow$ BFS($A, t, V$)
25:         $N \leftarrow N \cup n$
26:     **end for**
27:     **return** $N$
28: **end function**

---

## A.6 Wave-Pulse.io Frontend

Wave-Pulse.io is a comprehensive real-time data visualization platform that leverages React for the frontend and Django with PostgreSQL for the backend. The system is designed to create an intuitive and responsive user interface that facilitates data analysis and exploration.

The frontend, built with React, establishes communication with the backend through REST API calls, utilizing Axios for data management. To enhance performance and user experience, the frontend implements asynchronous data fetching techniques and employs caching mechanisms.

The platform comprises key components, each serving a specific purpose in the data visualization ecosystem:

*A.6.1 Home Page.* The Home Page functions as the central navigation hub for the Wave-Pulse.io application, providing users with access to various features and data visualization options.

*A.6.2 Map UI.* The Map UI is built upon the ComposableMap component from 'react-simple-maps', offering users an interactive exploration of the United States map. This visualization includes state boundaries, county outlines, and markers representing population centers and radio station coverage areas.

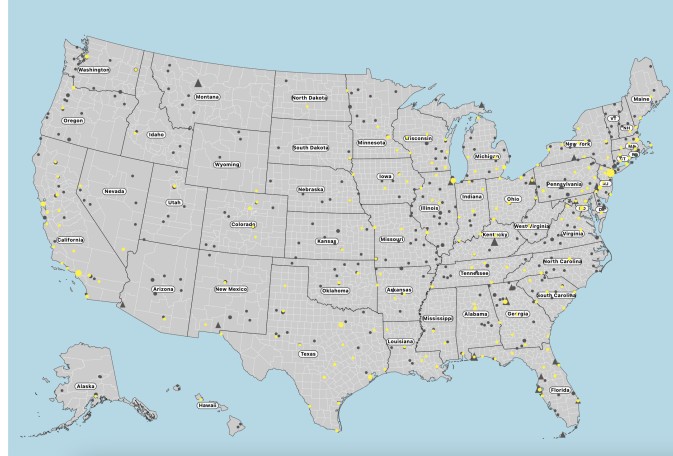

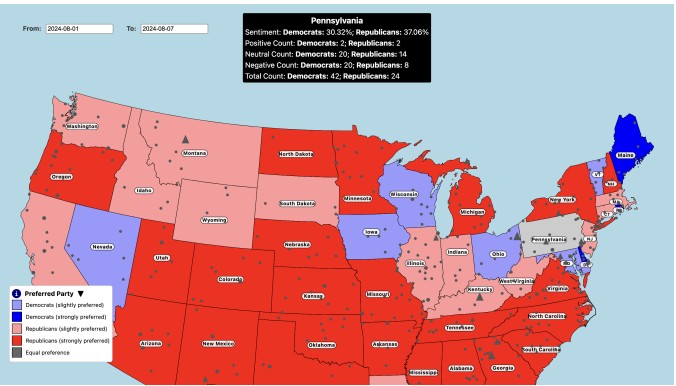

Figure 13: Wave-pulse USA Map illustrating state and county boundaries, with radio station markers (grey) and population centers (yellow) plotted according to their precise coordinates.

Figure 14: Detailed map visualization showing Republican vs Democrat leaning states from August 1, 2024, to August 7, 2024, with a particular emphasis on Pennsylvania data for the specified period.

To enhance user interaction and data analysis capabilities, the map interface offers toggles and filters:

*Map level toggle.*

- State Party Toggle: Visualizes sentiment data for political parties at the state level, providing insights into political leanings across regions.
- State Candidate Toggle: Presents sentiment data for individual candidates at the state level, allowing for comparison of candidate popularity.
- State Party Absolute Toggle: Displays absolute sentiment counts for political parties at the state level, offering a quantitative view of party support.
- State Candidate Absolute Toggle: Shows absolute sentiment counts for candidates at the state level, enabling direct numerical comparisons.

*Coverage toggle.*

- Show County Toggle: Activates county-level data visualization, allowing for granular analysis of sentiment patterns.
- Show Coverage Toggle: Illustrates the approximate radio area coverage for each station, providing insights into broadcast reach.
- Show Population Toggle: Highlights population density markers for major U.S. cities, contextualizing sentiment data with demographic information.

*Narrative toggle.* The 'Georgia Election Stolen' option provides a visualization of how information related to the 2020 Georgia election controversy propagated through the radio network, offering insights into information dissemination patterns.

*Date Picker.* This feature empowers users to select specific date ranges for data visualization, enabling temporal analysis of sentiment trends and patterns.

### A.6.3 Plots UI.
The Plots UI component features two primary line charts: a nationwide combined sentiment analysis and a nationwide sentiment count. These charts are designed to adjust based on user interactions, providing a responsive data exploration experience. To add depth to the visualizations, entropy is incorporated, providing context to the plotted data and illustrating the degree of uncertainty in the sentiment analysis.

*Sentiment Analysis.* This chart focuses on presenting sentiment values for key political entities (Biden, Harris, Trump, Democrats, and Republicans) on a daily basis. To smooth out short-term fluctuations and highlight longer-term trends, the system calculates and displays a 3-day moving average.

*Sentiment Count.* This visualization plots sentiment counts for both candidates and political parties. Users have the flexibility to toggle between different data lines, allowing them to focus on positive, neutral, or negative sentiments as needed for their analysis.

## A.7 Wave-Pulse.io Backend

The Django-powered backend is designed to expose APIs that facilitate interaction with the frontend. This backend infrastructure is responsible for processing, aggregating, and formatting data to enable real-time visualizations. The PostgreSQL database underpinning the system is optimized to handle complex queries and store time-series data, ensuring rapid data retrieval and analysis capabilities.

Wave-Pulse.io employs a hosting solution that leverages the strengths of multiple platforms. GitHub Pages is utilized to host the frontend, while the DigitalOcean App Platform is responsible for hosting the backend and PostgreSQL database. This setup ensures efficient resource utilization and cost-effectiveness, while maintaining high performance and scalability.

The deployment process is integrated with GitHub repositories, facilitating a Continuous Integration/Continuous Deployment (CI/CD) pipeline. This integration allows for updates and ensures that the latest stable version of the application is available to users.

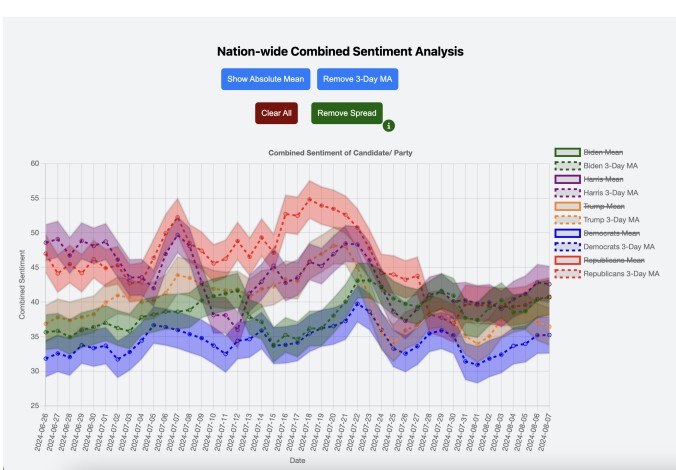

**Figure 15: Comprehensive sentiment plot for all parties and candidates, covering the period from June 26, 2024, to August 7, 2024.**

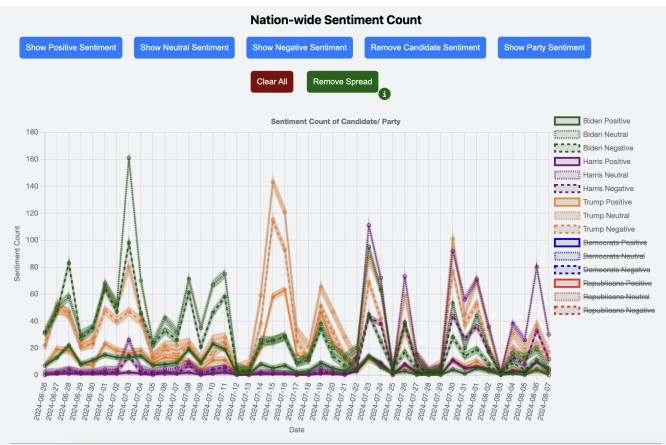

**Figure 16: Detailed plot illustrating the frequency of sentiment occurrences for all candidates over the period from June 26, 2024, to August 7, 2024.**

*A.7.1 GitHub Pages Frontend.* GitHub Pages serves as the hosting platform for the frontend, providing key advantages:

*Performance.* GitHub Pages delivers pre-built assets directly to the browser, resulting in fast load times for users. This approach eliminates the need for server-side rendering, enhancing the responsiveness of the application.

*Automated Deployment.* Updates to the frontend trigger automatic deployment processes using GitHub Actions. This automation streamlines the development workflow and ensures that new features and improvements are made available to users.

*Advantages.*

- Cost-effectiveness: GitHub Pages offers free hosting for public repositories, reducing operational costs.

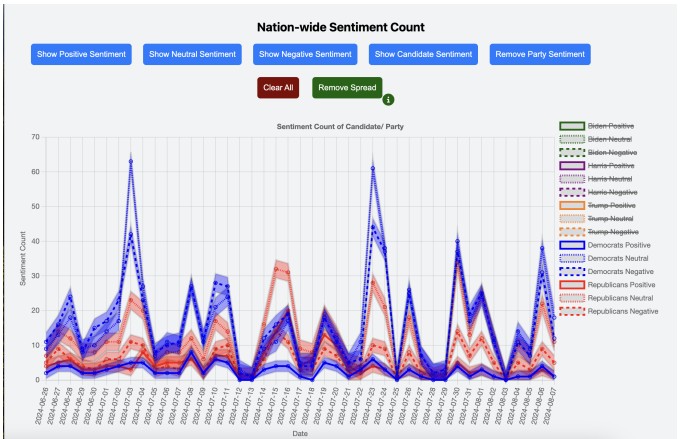

**Figure 17: Comprehensive plot showing the frequency of sentiment occurrences for both major political parties from June 26, 2024, to August 7, 2024.**

- Seamless integration: The tight integration with the GitHub ecosystem facilitates a smooth development and deployment process.
- Version control: Inherent version control capabilities allow for tracking of changes and rollbacks if necessary.

*A.7.2 DigitalOcean Backend.* The DigitalOcean App Platform is employed to handle the Django backend and PostgreSQL database, offering a robust and scalable solution for server-side operations.

*Functionality.* The backend processes incoming requests, manages complex business logic, and interfaces with the database to serve real-time data to the frontend. This setup ensures efficient data management and enables the dynamic features of the WavePulse.io platform.

*Advantages.*

- Dynamic resource allocation: DigitalOcean adjusts resource allocation based on the backend's workload, ensuring optimal performance during peak usage periods.
- Comprehensive monitoring: Integrated monitoring tools provide real-time insights into system performance, allowing for proactive management and optimization.
- Streamlined deployment: Automatic deployment processes are triggered by GitHub pushes, ensuring that the backend remains synchronized with the latest code changes.
- Managed database services: DigitalOcean's managed PostgreSQL service reduces the operational overhead of database management while maintaining high availability and performance.

## A.8 Useful Addition: Scraping Fact Checks

The fact-checking program is a system designed for the automated collection, processing, and post-processing of fact-checking articles from various reputable websites. Developed using Python and leveraging the Scrapy framework for web scraping, the system

incorporates specialized spiders for specific fact-checking websites, including FactCheck.org, Lead Stories, Politifact, Snopes, and TruthOrFiction. *Although we do not use it in WAVEPULSE, matching fact checks from authoritative resources with transcript embeddings can help automate fact-checking.*

*A.8.1 Websites Overview.* Our team developed a Fact Check Web Crawler capable of scraping fact-checking articles from multiple authoritative websites. The data collection period spans from January 1, 2020, to August 6, 2024, providing a substantial dataset for analysis. To ensure ethical and legal compliance, we reviewed the terms of service for each website prior to data collection.

*A.8.2 Scrapy Implementation.* The core of our fact-checking system utilizes Scrapy spiders to scrape fact-checking articles. These spiders are designed with flexibility, supporting various filtering options including date range, keywords, tags, and pagination. This adaptability allows for targeted data collection based on specific research needs.

Post-crawling, the system employs a data merging process that combines information from all scraped websites into a unified dataset. A key feature of our system is the standardization of fact-checking rulings across different articles, ensuring consistency in our analysis. The merged dataset undergoes a filtering process to isolate political content, enabling focused studies on political misinformation.

*A.8.3 Deduplication Process.* To ensure data integrity and prevent redundancy, we implemented a deduplication module. This component is designed to identify and manage duplicate fact-checking articles by conducting analysis of textual content and publication dates.

The deduplication process employs natural language processing techniques, including TF-IDF (Term Frequency-Inverse Document Frequency) vectorization. This method transforms the text into a numerical representation, enabling the calculation of cosine similarity between articles. By setting thresholds for similarity and considering publication date proximity, the system clusters and manages duplicate content.

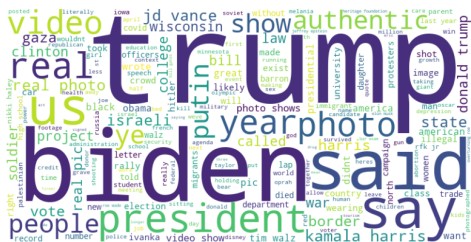

Figure 18: Illustrative example of misinformation fact-checked across multiple reputable websites, demonstrating the need for effective deduplication.

*A.8.4 Results Visualization.* To facilitate understanding and analysis of the collected fact-checking data, we developed visualization tools, including word clouds and histograms.

*Histogram Analysis.* We created a histogram to visualize the frequency distribution of various fact-checking rulings. This analysis provides insights into the landscape of misinformation and fact-checking efforts. For the year 2024, our findings include:

- The "False" category shows the highest frequency, with 1008 instances, indicating a significant volume of debunked claims.
- "True" claims occur less frequently, with 320 instances, suggesting a lower proportion of verified information in the fact-checking landscape.
- Categories such as "Miscaptioned", "Satire", and "Outdated" constitute a notable portion of the dataset, highlighting the diverse nature of misinformation and the nuanced approach required in fact-checking.

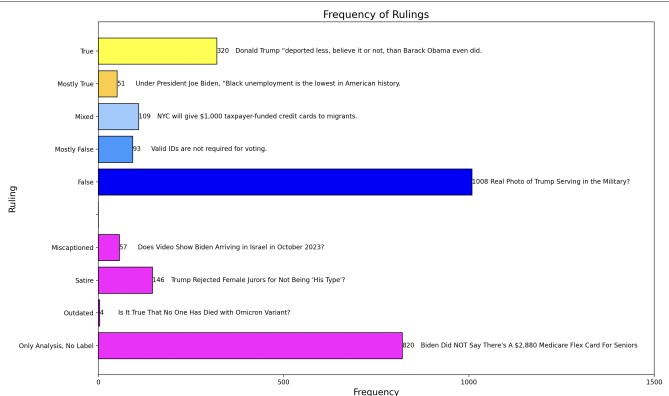

Figure 19: Comprehensive histogram illustrating the frequency of occurrence for all fact-checking "rulings" across the scraped websites for the year 2024.

Figure 20: Word cloud showing frequency of occurrence of significant words with the ruling "True" across all the data scraped from fact-checking websites for 2024.

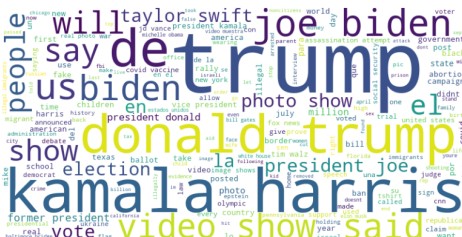

**Figure 21: Word cloud showing frequency of occurrence of significant words with the ruling "False" across all the data scraped from fact-checking websites for 2024.**

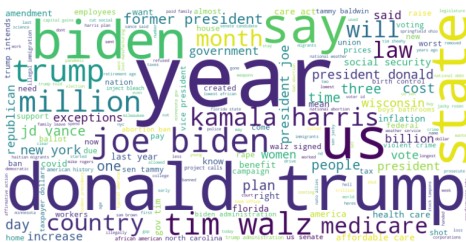

**Figure 22: Word cloud showing frequency of occurrence of significant words with the ruling "Mostly True", "Mixed", or "Mostly False" across all the data scraped from fact-checking websites for 2024.**

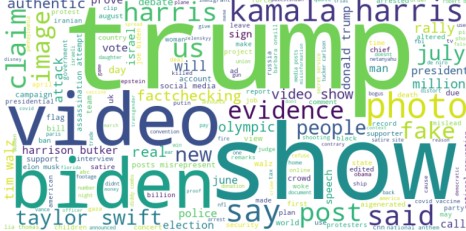

**Figure 23: Word cloud showing frequency of occurrence of significant words with the remaining rulings such as "Satire" or "Outdated" across all the data scraped from fact-checking websites for 2024.**

*Word Cloud Visualization.* To provide a representation of the fact-checking landscape, we generated word clouds that visually depict the most frequent words found in article titles, categorized by their fact-checking "rulings". This approach offers insights into recurring themes and topics within different truth categories. The titles were divided into four primary categories:

- True: Word clouds for this category often feature prominent political figures and terms related to electoral processes, such as "election", "president", and "vote".
- False: While containing similar political keywords, this category notably includes terms like "photo", "fake", and "real", indicating a prevalence of visual misinformation.
- Middle (including "Mostly True", "Mixed", or "Mostly False"): This category highlights words such as "year", "million", and "country", suggesting a focus on claims involving statistics or demographic information.
- Miscellaneous: Words like "video", "claim", "fact-checking", and "evidence" are more prominent in this category, reflecting the complex nature of these fact-checks.

## A.9 Useful Addition: Audio Classifier

As part of our data analysis toolkit, we developed a system for segmenting and classifying speech from audio files. This system leverages the audio processing capabilities of MediaPipe, enabling us to extract and analyze spoken content with accuracy. *Although we do not use it in WavePulse, we can use it to separate music from speech content.*

*Audio Segmentation.* The process begins with the loading of the audio file into our system. Once loaded, the audio data is converted into a numpy array, allowing for manipulation and analysis. The system then employs a segmentation approach, dividing the audio into fixed-length chunks with a specified overlap between consecutive segments. This segmentation is crucial for analyzing smaller portions of the audio stream, enabling us to capture specific speech events with precision.

The overlap between segments plays a role in ensuring continuity and preventing the loss of important speech elements that might occur at segment boundaries. For instance, in a scenario with a sample rate of 44.1 kHz, a one-second audio segment would contain 44,100 data points. If we set the segment length to 5 seconds, each segment would encompass 220,500 samples. With an overlap of 2 seconds, each segment would share 88,200 samples with its predecessor, ensuring smooth transitions and coverage.

*MediaPipe AudioClassifier Implementation.* Following the segmentation process, each audio chunk is passed through MediaPipe's AudioClassifier. This step begins with the loading of a pre-trained classification model, which we configure to return the top classification result for each segment, optimizing for accuracy and processing efficiency.

Prior to classification, each segment undergoes a normalization process. This involves dividing the audio values by the maximum possible value for 16-bit audio (32,767), ensuring that the input data falls within the appropriate range for the classifier. This normalization step is crucial for maintaining consistency and improving the accuracy of our speech detection.

The classifier evaluates each normalized segment, generating classification results that include both labels and confidence scores. We implement a quality control measure by considering a segment as containing valid speech only if the classifier assigns it a confidence score exceeding 0.80. This threshold helps minimize false positives and ensures the reliability of our speech detection.

*Final Audio File Compilation.* After processing all segments, our system combines the identified speech-containing segments into a single audio file. During this compilation, we remove the overlap between consecutive segments to avoid any duplication of audio data. This approach ensures that the final combined audio maintains continuity without repeating any part of the speech, resulting in a streamlined audio file containing only the relevant speech segments.

The resulting audio file, representing a distilled version of the original input focusing on detected speech, is then saved in the specified output directory. This final product serves as a resource for further analysis, transcription, or other downstream processing tasks.

**Table 2: List of successfully streamed Radio Stations along with their location.**

| Call Sign | Location | Call Sign | Location | Call Sign | Location |
|---|---|---|---|---|---|
| WACV | Coosada, AL | WAVH | Daphne, AL | WGSV | Guntersville, AL |
| WLBF | Montgomery, AL | WQSI | Union Springs, AL | WTLS | Tallassee, AL |
| KAGV | Big Lake, AK | KBKO | Kodiak, AK | KFAR | Fairbanks, AK |
| KFNP | North Pole, AK | KGSM | Saint Mary's, AK | KSRM | Soldotna, AK |
| KVNT | Eagle River, AK | KAWC | Yuma, AZ | KDJI | Holbrook, AZ |
| KFNN | Mesa, AZ | KFNX | Cave Creek, AZ | KQNA | Prescott Valley, AZ |
| KVOI | Cortaro, AZ | KVWM | Show Low, AZ | KYCA | Presott, AZ |
| KARV | Russellville, AR | KBEU | Bearden, AR | KBTM | Jonesboro, AR |
| KOMT | Lakeview, AR | KRZP | Gassville, AR | KUAR | Little Rock, AR |
| KURM | Rogers, AR | KAHI | Auburn, CA | KBLA | Santa Monica, CA |
| KCAA | Loma Linda, CA | KCNR | Shasta, CA | KINS | Blue Lake, CA |
| KMET | Banning, CA | KMYC | Marysville, CA | KOMY | La Selva Beach, CA |
| KPAY | Chico, CA | KPRL | Paso Robles, CA | KQMS | Redding, CA |
| KSAC | Olivehurst, CA | KSCO | Santa Cruz, CA | KVTA | Ventura, CA |
| KYOS | Merced, CA | KDGO | Durango, CO | KFKA | Greeley, CO |
| KGLN | Glenwood Springs, CO | KLZ | Denver, CO | KNFO | Basalt, CO |
| KPPF | Monument, CO | KRDO | Colorado Springs, CO | KVFC | Cortez, CO |
| WDRC | Hartford, CT | WFOX | Southport, CT | WGCH | Greenwich, CT |
| WICC | Bridgeport, CT | WLAD | Danbury, CT | WSTC | Stamford, CT |
| WDEL | Wilmington, DE | WGMD | Reho. Beach, DE | WHMS | Pine Creek, DE |
| WIHW | Dover, DE | WVCW | Wilmington, DE | WCSP | Washington, DC |
| WFED | Washington, DC | WPFM | Washington, DC | WTOP | Washington, DC |
| PRNN | Pensacola, FL | WBOB | Jacksonville, FL | WDBO | Orlando, FL |
| WDCF | Dade City, FL | WELE | Ormond Beach, FL | WFSX | Estero, FL |
| WFTL | West Palm Beach, FL | WHBO | Pinellas Park, FL | WKEZ | Tavernier, FL |
| WNDB | Daytona Beach, FL | WNRP | Pensacola, FL | WNZF | Bunnell, FL |
| WPIK | Summerland Key, FL | WPSL | Port Saint Lucie, FL | WWBA | Largo, FL |
| WWPR | Bradenton, FL | WWTK | Lake Placid, FL | WXJB | Homosassa, FL |
| WYOO | Springfield, FL | WCHM | Clarkesville, GA | WDJY | Dallas, GA |
| WDUN | Gainesville, GA | WFOM | Marietta, GA | WGAC | Harlem, GA |
| WJRB | Young Harris, GA | WKWN | Trenton, GA | WLAQ | Rome, GA |
| WLBB | CaWAUBrrollton, GA | WRGA | Rome, GA | WRWH | Cleveland, GA |
| WSBB | Doraville, GA | WVGA | Lakeland, GA | WVOP | Vidalia, GA |
| KANO | Hilo, HI | KHJC | Lihue, HI | KIHL | Hilo, HI |
| KKCR | Hanalei, HI | KAOX | Shelley, ID | KBOI | New Plymouth, ID |
| KIDG | Shelley, ID | KOUW | Island Park, ID | WBGZ | Alton, IL |
| WCGO | Evanston, IL | WCIL | Carbondale, IL | WCMY | Ottawa, IL |
| WCPT | Willow Springs, IL | WCRA | Effingham, IL | WDAN | Danville, IL |
| WDWS | Champaign, IL | WGGH | Marion, IL | WJPF | Herrin, IL |
| WLUW | Chicago, IL | WMAY | Taylorville, IL | WMBD | Peoria, IL |
| WRPW | Colfax, IL | WSDR | Sterling, IL | WSOY | Decatur, IL |
| WTAD | Quincy, IL | WTIM | Assumption, IL | WTRH | Ramsey, IL |
| WZUS | Macon, IL | WBIW | Bedford, IN | WFDM | Franklin, IN |
| WGCL | Bloomington, IN | WGL | Fort Wayne, IN | WIMS | Michigan City, IN |
| WTRC | Elkhart, IN | KBIZ | Ottumwa, IA | KFJB | Marshaltown, IA |
| KMA | Shenandoah, IA | KOKX | Keokuk, IA | KWBG | Boone, IA |
| KXEL | Waterloo, IA | KGGF | Coffeyville, KS | KINA | Salina, KS |
| KIUL | Garden City, KS | KLWN | Lawrence, KS | KQAM | Wichita, KS |
| KSAL | Salina, KS | KSCB | Liberal, KS | KVGB | Great Bend, KS |
| KWBW | Hutchinson, KS | KWKN | Wakeeney, KS | WDOC | Prestonsburg, KY |
| WHIR | Danville, KY | WKCT | Bowling Green, KY | WZXI | Lancaster, KY |
| KFXZ | Lafayette, LA | KSYL | Alexandria, LA | KWLA | Anacoco, LA |

**Table 3: List of successfully streamed Radio Stations along with their location (continued).**

| Call Sign | Location | Call Sign | Location | Call Sign | Location |
|---|---|---|---|---|---|
| WBOK | New Orleans, LA | WEGP | Presque Isle, ME | WLOB | Portland, ME |
| WMEA | Portland, ME | WBAL | Baltimore, MD | WCBM | Baltimore, MD |
| WFMD | Frederick, MD | WBNW | Concord, MA | WGAW | Gardner, MA |
| WNBP | Newburyport, MA | WSAR | Fall River, MA | WAAM | Ann Arbor, MI |
| WBRN | Big Rapids, MI | WCXI | Fenton, MI | WIOS | Tawas City, MI |
| WKHM | Jackson, MI | WKNW | Sault Sainte Marie, MI | WLDN | Ludington, MI |
| WMIC | Sandusky, MI | WMPL | Hancock, MI | WPHM | Port Huron, MI |
| WSJM | Benton Harbor, MI | WTCM | Traverse City, MI | KBRF | Fergus Falls, MN |
| KKBJ | Bemidji, MN | KLTF | Little Falls, MN | KNSI | Saint Louis, MN |
| KROX | Crookston, MN | KTRF | Thief River Falls, MN | KXRA | Alexandria, MN |
| WZFG | Dilworth, MN | WMXI | Ellisville, MS | WVBG | Vicksburg, MS |
| WYAB | Pocahontas, MS | KFMO | Flat River, MO | KICK | Springfield, MO |
| KRMS | Osage Beach, MO | KRTK | Hermann, MO | KSIM | Sikeston, MO |
| KSWM | Aurora, MO | KTRS | Saint Louis, MO | KTTR | Saint James, MO |
| KTUI | Sullivan, MO | KWOC | Poplar Bluff, MO | KWPM | West Plains, MO |
| KZIM | Cape Girardeau, MO | KZRG | Joplin, MO | KZYM | Joplin, MO |
| KAFH | Great Falls, MT | KALS | Kalispell, MT | KAPC | Butte, MT |
| KBGA | Missoula, MT | KBMC | Bozeman, MT | KCAP | Helena, MT |
| KINX | Fairfield, MT | KJJR | Whitefish, MT | KGFW | Kearney, NE |
| KLIN | Lincoln, NE | KODY | North Platte, NE | KOIL | Omaha, NE |
| KOLT | Terrytown, NE | KRGI | Grand Island, NE | WJAG | Norfolk, NE |
| KAVB | Hawthorne, NV | KELY | Ely, NV | KKFT | Gardnerville-Minden, NV |
| KLNR | Panaca, NV | KNCC | Elko, NV | WEMJ | Laconia, NH |
| WNTK | New London, NH | WTSN | Dover, NH | WUVR | Lebanon, NH |
| WFJS | Trenton, NJ | WFMU | East Orange, NJ | WOND | Pleasantville, NJ |
| WVBV | Medford Lakes, NJ | KEND | Roswell, NM | KENN | Farmington, NM |
| KINN | Alamogordo, NM | KKOB | Albuquerque, NM | KOBE | Las Cruces, NM |
| KRSY | Alamogordo, NM | KSVP | Artesia, NM | KXKS | Albuquerque, NM |
| WATN | Watertown, NY | WAUB | Auburn, NY | WBAI | New York, NY |
| WFME | Garden City, NY | WGBB | Freeport, NY | WGDJ | Rensselaer, NY |
| WGVA | Geneva, NY | WJJF | Montauk, NY | WKCR | New York, NY |
| WLNL | Horseheads, NY | WLVL | Lockport, NY | WNYU | New York, NY |
| WRHU | Hempstead, NY | WTBQ | Warwick, NY | WUTQ | Utica, NY |
| WVBN | Bronxville, NY | WWSK | Smithtown, NY | WYSL | Avon, NY |
| WBT | Charlotte, NC | WEEB | Southern Pines, NC | WGNC | Gastonia, NC |
| WHKY | Hickory, NC | WNOS | New Bern, NC | WOBX | Wanchese, NC |
| WRHT | Morehead City, NC | WSJS | Winston-Salem, NC | WSPC | Albemarle, NC |
| WTIB | Willamston, NC | KNOX | Grand Forks, ND | KTGO | Tioga, ND |
| WDAY | Fargo, ND | WCBE | Columbus, OH | WDBZ | Cincinnati, OH |
| WHIO | Dayton, OH | WHTX | Warren, OH | WINT | Willoughby, OH |
| WLYV | Bellaire, OH | WNIR | Kent, OH | WYOH | Niles, OH |
| KCLI | Cordell, OK | KGWA | Enid, OK | KGYN | Guymon, OK |
| KQOB | Enid, OK | KRMG | Tulsa, OK | KTLR | Oklahoma City, OK |
| KWON | Bartlesville, OK | WBBZ | Ponca City, OK | KAGO | Klamath Falls, OR |
| KBND | Bend, OR | KBNP | Portland, OR | KFIR | Sweet Home, OR |
| KFLS | Klamath Falls, OR | KGAL | Lebanon, OR | KMED | Eagle Point, OR |
| KPNW | Eugene, OR | KSLM | Salem, OR | KUMA | Pendleton, OR |
| KVBL | Union, OR | KWRO | Coquille, OR | KYKN | Keizer, OR |
| WATS | Sayre, PA | WBVP | Beaver Falls, PA | WCED | Du Bois, PA |
| WEEU | Reading, PA | WFYL | King of Prussia, PA | WKHB | Irwin, PA |
| WPSN | Honesdale, PA | WRSC | Bellefonte, PA | WRTA | Altoona, PA |

**Table 4: List of successfully streamed Radio Stations along with their location (continued).**

| Call Sign | Location | Call Sign | Location | Call Sign | Location |
|---|---|---|---|---|---|
| WTRW | Carbondale, PA | WURD | Philadelphia, PA | WEAN | Wakefield-Peacedale, RI |
| WNPE | Narragansett Pier, RI | WSJW | Pawtucket, RI | WAIM | Anderson, SC |
| WCRS | Greenwoord, SC | WDXY | Sumter, SC | WFRK | Quinby, SC |
| WRHI | Rock Hill, SC | WRNN | Socastee, SC | WTKN | Murrells Inlet, SC |
| KAUR | Sioux Falls, SD | KELQ | Flandreau, SD | KOTA | Rapid City, SD |
| KWAM | Memphis, TN | WBFG | Parker's Crossroads, TN | WCMT | Martin, TN |
| WENO | Nashville, TN | WGNS | Murfreesboro, TN | WHUB | Cookeville, TN |
| WUCT | Algood, TN | KBST | Big Spring, TX | KCRS | Midland, TX |
| KKSA | San Angelo, TX | KLVT | Levelland, TX | KRDY | San Antonio, TX |
| KRFE | Lubbock, TX | KWEL | Midland, TX | KXYL | Brownwood, TX |
| KZHN | Paris, TX | KZHN | Paris, TX | WTAW | College Station, TX |
| KBJA | Sandy, UT | KJJC | Murray, UT | KMXD | Monroe, UT |
| KOAL | Price, UT | KSGO | Saint George, UT | KSVC | Richfield, UT |
| KVNU | Logan, UT | WBTN | Bennington, VT | WCKJ | Saint Johnsbury, VT |
| WJPL | Barre, VT | WJSY | Newport, VT | WMTZ | Rutland, VT |
| WVMT | Burlington, VT | WCHV | Charlottesville, VA | WFJX | Roanake, VA |
| WGMN | Roanake, VA | WIQO | Forest, VA | WJFV | Portsmouth, VA |
| WLNI | Lynchburg, VA | WMNA | Gretna, VA | WNIS | Norfolk, VA |
| WRAD | Radford, VA | WRCW | Warrenton, VA | KEDO | Longview, WA |
| KELA | Centralia-Chehalis, WA | KGDC | Walla Walla, WA | KGTK | Olympia, WA |
| KITZ | Silverdale, WA | KKNW | Seattle, WA | KLCK | Goldendale, WA |
| KNWN | Seattle, WA | KODX | Seattle, WA | KONP | Port Angeles, WA |
| KOZI | Chelan, WA | KSBN | Spokane, WA | KTEL | Walla Walla, WA |
| KVI | Seattle, WA | KXLY | Spokane, WA | WMOV | Ravenswood, WV |
| WRNR | Martinsburg, WV | WSCW | South Charleston, WV | WWNR | Beckley, WV |
| KFIZ | Fond Du Lac, WI | WAUK | Jackson, WI | WCLO | Janesville, WI |
| WFHR | Wisconsin Rapids, WI | WISS | Berlin, WI | WLCX | La Crosse, WI |
| WMDX | Columbus, WI | WSAU | Rudolph, WI | WTAQ | Glenmore, WI |
| WXCO | Wausau, WI | KBUW | Buffalo, WY | KROE | Sheridan, WY |
| KVOW | Riverton, WY | | | | |

