# OpenReview forum: "WavePulse: Real-time Content Analytics of Radio Livestreams"
_ACM.org/TheWebConf/2025/Conference — WWW 2025 Poster_

### Official Review · Reviewer_Y9nW · 2024-11-22

**Novelty:** 5
**Technical Quality:** 5

**Review:**

The paper introduces a system for monitoring and analysis of radio broadcasts. It describes the deployment of a framework for such analysis and 3 case studies that focus on the reporting and discussion of US election related topics on US radio stations streamed online over a period of a few months. The 3 case studies are on content analysis over the transcripts that were collected and their findings are discussed.

The validity of the findings on favorability of election candidates in the third case study is argued by a comparison to those in opinion polls. The first case study is on how political narratives are reported and there is some discussion and comparison to manual selection of transcripts versus LLM-based selection; the LLM based selection provided many more samples for such assessment. The second case study demonstrates how content syndication was monitored based on the collected datasets.

This is a relevant paper that would generate valuable discussion at the conference and its contribution lies both on the architecture and the methodology. It is well-written and the case for radio transcript generation and assessment is well argued.

**Questions:**

The paper could be strengthened by discussing in a little more detail the validity of the findings of the first and the second case studies. In the first case study indeed the LLM approach provided many more samples but it is not entirely clear whether those samples were indeed accurately selected or whether the manual selection could be improved. Similarly, in the second case study there could be more discussion on the validity of the findings on content syndication.

**Reviewer Confidence:**

4: The reviewer is certain that the evaluation is correct and very familiar with the relevant literature

**Scope:**

4: The work is relevant to the Web and to the track, and is of broad interest to the community

---

### Official Review · Reviewer_TRMu · 2024-12-01

**Novelty:** 6
**Technical Quality:** 5

**Review:**

> Quality

The paper presents a comprehensive framework for capturing, analyzing, and understanding content from radio livestreams, particularly in the context of political discourse during the 2024 Presidential Elections. The quality of the work is high, as evidenced by the sophisticated use of AI tools, including large language models, for real-time acquisition, transcription, speaker diarization, and content analysis. The framework's ability to process a massive amount of data, converting it into actionable insights, demonstrates a high level of technical proficiency.

> Clarity

The paper is well-structured and clearly articulated. The abstract effectively summarizes the key points, and the introduction provides a strong motivation for the work. The methodology is explained in a step-by-step manner, making it easy to follow. The results are presented clearly, with figures and tables that effectively illustrate the findings. The case studies are particularly well-explained, providing concrete examples of how the WavePulse framework can be applied in real-world scenarios.

> Originality

The originality of the work lies in its end-to-end approach to radio content analytics. While there has been prior work in radio content transcription and analysis, WavePulse stands out by offering a complete pipeline that not only transcribes and analyzes content but also provides real-time insights at scale. The use of modern multimodal large language models (LLMs) for this purpose is a novel contribution.

> Significance

The significance of this work is underscored by its potential impact on political science research, media analysis, and public discourse. By providing a detailed analysis of radio content, WavePulse offers a unique perspective on information flow and public sentiment during a critical period. This could inform policy, media literacy efforts, and academic research.

> Pros

- **End-to-End Pipeline:** The comprehensive nature of the framework covering all stages from data acquisition to analysis is a significant advantage.
- **Real-Time Analytics:** The capability for real-time processing and analysis is particularly valuable for dynamic environments like political campaigns.
- **Large Scale Data Handling:** Processing nearly half a million hours of content is impressive and speaks to the scalability of the framework.
- **Application in Political Science:** The case studies demonstrate the practical utility of the framework in understanding political discourse and public sentiment.

>  Cons

- **Data Public Availability:** One of the critical questions regarding this paper is the **public availability of the data used in the study**. Open data sharing is essential for reproducibility, validation of results, and extension of research by other scientific communities.
  - 1. **Data Accessibility:** Will the dataset generated from the 485,090 hours of speech content be made publicly available? If so, could
you detail the conditions under which it will be shared (e.g., time frame for release, licensing terms)?

  - 2. **Data Usage Rights:** Are there any restrictions on the use of the data, especially considering it contains politically sensitive content? How will you handle issues related to copyright, especially with syndicated content?

  - 3. **Anonymization:** How was the data anonymized to protect the privacy of individuals mentioned in the broadcasts? Could you provide details on the methods used to ensure no personally identifiable information (PII) is disclosed?

  - 4. **Data Format and Documentation:** In what format will the data be released, and will there be accompanying documentation to assist researchers in understanding and utilizing the dataset effectively?

- **Model and Code Availability:** The accessibility of the models and code used in the WavePulse framework is another significant aspect for the research community, particularly for those interested in building upon or integrating this technology into their work.

  - 1. **Model Availability:** Are the AI models used in the WavePulse framework, particularly the large language models for transcription and analysis, available for public use? If not, are there plans to release them or similar models under an open-source license?

  - 2. **Code Sharing:** Will the codebase of the WavePulse framework be open-sourced? If so, when can the community expect it to be available, and what will be the licensing terms?

  - 3. **Documentation and Tutorials:** Will documentation and tutorials accompany the release of the code to facilitate its use by other researchers and developers? If yes, could you provide an overview of what will be included in these resources?

  - 4. **Dependencies and Reproducibility:** What are the dependencies required to run the WavePulse framework, and have you taken steps to ensure the reproducibility of your results, such as providing Docker containers or virtual environments with all necessary libraries and versions specified?

  - 5. **Updates and Maintenance:** Who will be responsible for maintaining the code and models post-release, and is there a plan for regular updates to accommodate new research findings or technological advancements?

- **Generalizability:** While the framework is applied to political radio content, it is unclear how well it would perform with non-political content or in other languages.
- **Bias and Ethical Considerations:** The paper could benefit from a deeper discussion on potential biases in the AI models used and ethical considerations regarding data privacy and consent.

**Questions:**

1. **Generalizability Across Content Types:** Can the WavePulse framework be effectively applied to non-political radio content? If so, have there been any tests or considerations regarding its performance in such scenarios?

2. **Multilingual Support:** Are there plans to extend the framework to support multiple languages, given the global nature of radio broadcasting?

3. **Bias in AI Models:** How do you address and mitigate potential biases in the AI models used within the framework, particularly in the context of political content analysis?

4. **Data Privacy:** What measures have been taken to ensure data privacy, especially with regards to the handling of personally identifiable information (PII) within the transcripts?

5. **Ethical Considerations:** Could you elaborate on the ethical considerations that guided the development and application of WavePulse, especially in the realm of political discourse analysis?

6. **Scalability Challenges:** What are the main challenges in scaling the framework to handle an even larger volume of data, and how do you propose to address them?

7. **Future Work:** Are there any plans to integrate WavePulse with social media analytics to provide a more holistic view of public discourse across different media platforms?

**Ethics Review Description:**

Data privacy issues and Bias in LLM Models

**Ethics Review Flag:**

Yes

**Reviewer Confidence:**

3: The reviewer is confident but not certain that the evaluation is correct

**Scope:**

3: The work is somewhat relevant to the Web and to the track, and is of narrow interest to a sub-community

---

### Official Review · Reviewer_iy1e · 2024-12-02

**Novelty:** 4
**Technical Quality:** 4

**Review:**

The paper presents WavePulse, a framework to record radio (online) transmissions in real-time. The framework is defined as an orchestration of APIs/models. The tool transcribes the recordings and performs some (standard) analyses over the transcriptions. The capabilities of the tool are exemplified by a case-study related to the US 2024 presidential elections. The idea of transcribing web radio is not exactly new, as there have been previous papers in the literature proposing tools for the task (e.g. “Web Radio Automation for Audio Stream Management in the Era of Big Data”), also including Speaker Diarization techniques. On the other hand, the data analysis techniques included are also from the literature.

Although the paper seems to fit the topics of the track (as it includes different types of the listed analysis), the contribution is more oriented towards the framework and the datasets, and the presented case studies are anectodical. Then, given the aim of the paper, in my opinion, the paper would be better suited for a tool/demo track, or even a resource track.


-- Introduction
* “While these distinctions make radio unique as a medium, they also make radio content much more challenging to monitor”. Why?

-- Framework and data collection.

---- A dataset of nationwide radio transcripts

* “This step also enhances suitability of our dataset for open research by filtering out personally identifiable information” What information needs to be filtered out?


-- Analysis and case studies

---- 3.1 Case study: Spread of a political narrative.

* “We collaborated with a democracy group at a non-profit center which champions social causes including election integrity” I understand that not too many details might not be disclosed due to the anonymity requirements, but if the goal of the collaboration was to “understand how WavePulse could be useful to gain insights”, at least the background of the people using the tool could have been included.
* Later on, the analysis of the case-studies are presented, but there’s no mentioned of the democracy group or their impressions when using the tools.
* There’s a list of keywords used for filtering the transcripts. Why use a lexicon-based search and not semantic similarity based on the available embeddings?

---- 3.2. Case Study: Content syndication across radio stations.
* It is stated that the authors developed an algorithm to identify unique broadcasts and their repeats. This is the first time this contribution is mentioned.
* The definition of the algorithm and its evaluation are not exhaustive.
---- 3.3. Case study: presidential candidates’ favorability trends
* It is stated that the authors developed a normalized sentiment score. To claim to have developed a normalized score, an evaluation of the differences between the proposed score and others in the literature should have been included.
* Figure 6. It’s not clear why the sentiment seems to move between 0 and 100, when the developed score ranges between 0 and 1.

-- Appendixes
* They are longer than the paper.
---- A.8. Useful addition: Scraping fact checks.
* Not clear the point of including this section, as it is not part of the proposal.
---- A.9. Useful addition: Audio classifier.
* “As part of our data analysis toolkit, we developed a system for segmenting and classifying speech from audio files. This system leverages the audio processing capabilities of MediaPipe, enabling us to extract and analyze spoken content with accuracy. Although we do not use it in WavePulse, we can use it to separate music from speech content.” Again, not clear the point of including this section.

-- Minor comments.
* Check typos. For example, in the last box in section 3.1.

**Questions:**

See above.

**Reviewer Confidence:**

4: The reviewer is certain that the evaluation is correct and very familiar with the relevant literature

**Scope:**

3: The work is somewhat relevant to the Web and to the track, and is of narrow interest to a sub-community

---

### Official Review · Reviewer_Jc55 · 2024-12-02

**Novelty:** 5
**Technical Quality:** 4

**Review:**

**Summary:**
This paper introduces WavePulse, a real-time framework for monitoring and analyzing radio livestreams, with a specific application to political science research during the 2024 Presidential Elections. The authors collect and transcribe a large corpus of radio content from 396 news radio stations, converting the audio into time-stamped, diarized transcripts. They showcase the utility of WavePulse by answering key political science questions and analyzing how local issues influenced national political trends. The results demonstrate the effectiveness of WavePulse in large-scale, real-time content analytics of radio livestreams.

**Strengths:**

1. **Comprehensive Experimentation and Dataset**:
   - One of the most significant strengths of the paper is the extensive and well-documented experimental setup. The authors processed nearly 500,000 hours of radio livestreams, making the dataset both large and diverse. The choice of monitoring 396 radio stations over three months allows for a thorough examination of various political trends at both national and state levels.
   - The paper provides a clear description of the methodologies used for transcribing and diarizing the audio streams, along with the challenges faced and the techniques employed to address them. This level of detail strengthens the paper's credibility and contributes to the transparency of the analysis.

2. **Real-World Application and Impact**:
   - The authors demonstrate WavePulse's practical utility by using it to track political trends during a high-profile event like the 2024 Presidential Elections. This not only showcases the tool's relevance but also highlights the broader applicability of the framework for real-time content analysis in domains such as politics, media studies, and public opinion research.
   - The findings of how local issues interact with national trends provide valuable insights for political scientists, further cementing the importance of the framework.

3. **Innovative Use of Technology**:
   - The integration of large language models (LLMs) for content analysis is another notable strength. The use of cutting-edge technology to transcribe and analyze large volumes of audio content in real-time represents an important advancement in the field of web content analytics.
   - The time-stamped, diarized transcription method is well-executed and crucial for enabling effective analysis of radio content, particularly for extracting political and social insights from real-time data.

**Weaknesses:**

1. **Limited Discussion of Methodological Challenges**:
   - While the paper covers the technical aspects of the framework and its application, there is insufficient discussion about the challenges associated with diarization and transcription of such a large and diverse dataset. Given the complexity of handling audio streams from various radio stations, it would be beneficial for the authors to elaborate on how the system handled issues like speaker identification errors, accents, or noisy audio.
   - A more detailed discussion of the limitations of the transcription and diarization models, and how they may impact the accuracy of the results, would help address potential concerns from readers.

2. **Comparative Analysis with Existing Tools**:
   - The paper does not provide enough context about how WavePulse compares with other real-time content analytics frameworks or radio monitoring tools. While the authors demonstrate its effectiveness in political content analysis, a comparison with existing systems would help highlight the unique advantages of WavePulse in terms of scalability, accuracy, and efficiency.
   - A more thorough review of related work in the domain of real-time content analytics, particularly for audio data, would improve the paper’s positioning within the broader research landscape.

3. **Potential Bias and Generalization of Results**:
   - Although the paper shows WavePulse’s efficacy in political content analysis, there is limited exploration of potential biases in the dataset. For example, the selection of radio stations may not fully represent the diversity of political discourse in the U.S., and the results might be skewed towards specific political perspectives or regional issues.
   - Future work should consider expanding the scope of the dataset to include a broader range of stations or diversify the geographic representation of the radio sources to ensure more balanced insights.

4. **Lack of Quantitative Evaluation Metrics**:
   - The paper provides a general qualitative assessment of the framework’s effectiveness but lacks specific quantitative metrics to evaluate its performance. For example, there are no metrics for the accuracy of the transcription, diarization, or content analysis models. Including such metrics (e.g., transcription accuracy, error rates, or analysis precision) would help quantify the framework's success and provide a clearer benchmark for future comparisons.

**Questions:**

See in weakness

**Reviewer Confidence:**

2: The reviewer is willing to defend the evaluation, but it is likely that the reviewer did not understand parts of the paper

**Scope:**

4: The work is relevant to the Web and to the track, and is of broad interest to the community

---

### Official Review · Reviewer_6yVa · 2024-12-02

**Novelty:** 4
**Technical Quality:** 5

**Review:**

This work contributes with (i) a framework for recording, and structuring web radio audio streams in real-time, and (ii) a demonstration of  how this collected data can be leveraged to track the propagation or dissemination of different societal opinions, including  politics with the case of the US 2024 presidential elections. From the analysis of 3 months data collected over the summer 2024 from 396 web radio stations, the authors successfully illustrated numerous case studies where they identified and tracked specific narratives and its propagation accross stations; clustered stations into those that mirror one another content, or measured over-the-wave potential political trends and showed that they aligned with general public opinions reported by specialized bodies like polling institutes.


## Pros
- Introduced WavePulse, a comprehensive LLM-based end-to-end framework for capturing, and storing real-time web radio streams into structured and queriedable data. By levearing different AI systems, the framework can tell apart different types of content, including political (news and discussions), apolitical content, advertisements, etc.
- Provided and tested different analysis pipelines, including topic modeling and sentiment analysis, to track the dissemination of news or the public sentiments regarding different narratives and persons.


## Cons
- The best way to query the data is not very clear. It seems that the QA-RAG method, described at the end of Section 2.2., gives the best results, but manual rules have also been employed in Section 3.1.
- The performance overhead of the framework, in terms of computation and storage is not reported.

**Questions:**

- Can you give examples of PII (Personnally Identifiable Information) that you can automatically remove from the dataset? In general, why are you worried about PII given that the radio dataset is public?
- What is the best way to query the dataset and can you give some hints on the (in)effectiveness of the manual rule-based system?
- Would it be possible to hook into WavePulse, in particular right after the JSON segments are generated, to plug-in a different storage mechanism (e.g., MongoDB) and query the different documents with the underlying query language of the custom storage mechanism?
- How does WavePulse perform, in real setup, in terms of computations, storages, etc.?

**Reviewer Confidence:**

3: The reviewer is confident but not certain that the evaluation is correct

**Scope:**

3: The work is somewhat relevant to the Web and to the track, and is of narrow interest to a sub-community